# ON THE RELATION BETWEEN GRADIENT DIRECTIONS AND SYSTEMATIC GENERALIZATION

## ABSTRACT

Systematic generalization is a critical property that most general deep learning algorithms lack. In this paper, we investigate the relation between gradient directions and systematic generalization. We propose a formulation to treat reducible training loss as a resource, and the training process consumes it to reduce test loss. We derive a bias that a training gradient is less efficient in using the resource at each step than an alternative gradient that leads to systematic generalization. The bias is avoided if and only if both gradients are zero or point in the same direction. We demonstrate the bias in standard deep learning models, including fully connected, convolutional, residual networks, LSTMs, and (Vision) Transformers. We also discuss a requirement for the generalization. We hope this study provides novel insights for improving systematic generalization. Source codes are available in the supplementary material.

## 1 INTRODUCTION

Deep learning has made remarkable progress across various domains, spanning from natural language processing to computer vision. Nevertheless, one of the biggest challenges in deep learning is to generalize the model to out-of-distribution (o.o.d.) data, which have zero probability in the training distribution. Systematic generalization (Fodor & Pylyshyn, 1988; Lake & Baroni, 2018) is a type of o.o.d generalization. It usually requires that a sample has multiple explanatory factors of variation (Bengio et al., 2013), and the generalization can produce an unseen combination of seen factor values. For example, models trained on blue rectangles and green triangles predict blue triangles. Systematic generalization is crucial for human learning and supports efficient data use and creativity. We hope machines acquire such generalization ability to achieve human-like intelligence.

The advantages of deep learning include its performance and generality. Many deep learning models achieve high accuracy on i.i.d. problems, though they often do not perform well on systematic generalization, as reported in recent studies (Hendrycks & Dietterich, 2019; Goyal et al., 2021b). Also, deep learning does not require many task-specific designs for specific tasks. Some standard networks, such as ResNets and Transformers, generally work well in i.i.d. settings. To keep the advantage, we discuss whether standard deep learning models achieve systematic generalization.

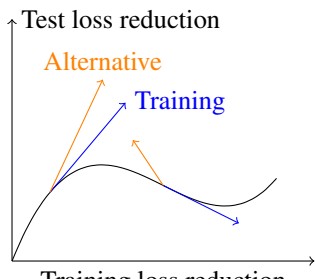

Figure 1: Illustration of training and test loss reductions. The horizontal axis is the training loss reduction. The vertical axis is the test loss reduction. A gradient changes both training and test losses, expressed as an arrow. An arrow for a training gradient points less upward in direction at different places than an alternative gradient that leads to systematic generalization.

If all factor combinations are available in training, a model is likely to work on the test combinations, because they are included in training. It means the generalization problem can be regarded as whether the training data can achieve the effect of all data. So we compare cases with training data and all data. Also, gradient descent is a common training algorithm in deep learning. So we study systematic generalization from a gradient perspective. We have a training gradient from training data and an

alternative gradient from all data. We focus on how the difference in gradient directions reduces generalization ability. If we assume the gradients have different directions in general deep learning models, then these models are not likely to achieve systematic generalization.

We introduce the concept of treating reducible training loss as a resource, which is consumed during the training process to reduce the test loss. We show that there is a bias in the training gradient, which reduces its efficiency in using this resource compared to the alternative gradient that leads to systematic generalization (please refer to Figure 1). The bias is only avoided when both gradients are zero or point in the same direction. It is supported by the derived Theorem 1.

We run experiments on standard deep learning models, including fully connected, convolutional, residual networks, LSTMs, and (Vision) Transformers. We demonstrate the existence of the bias in these models. We also discuss that systematic generalization requires a network decomposed to sub-networks, each with a seen test inputs.

In this paper, by investigating the relation between gradient directions and systematic generalization and characterizing the bias in the training gradient, we provide a new perspective on the problem. We hope our findings will inspire new research directions for improving systematic generalization in deep learning, ultimately leading to more robust and reliable models. The contributions can be summarized as follows.

- The main contribution is the proposed formulation: treat training loss as a resource; study the efficiency of the gradients; and propose a metric DDR to evaluate the efficiency.

- Based on the formulation, we derive the relation between gradients and generalization.

- Experiments validate the result and demonstrate a bias in standard deep learning models.

- We discuss a requirement for systematic generalization, based on which we explain why standard models do not achieve generalization and analyze existing approaches.

## 2 DEFINITIONS AND DERIVATIONS

We propose a framework to study the relationship between gradient directions and generalization. The main idea is to compare a training gradient and an alternative gradient. The alternative gradient is computed from all data, leading to systematic generalization. We look at the efficiency of gradients to reduce test loss while reducing training loss.

We treat the reducible training loss as a fixed amount of resources. We assume the final training losses have similar values when a model is trained well from the same initial parameters, which means the reduced losses are similar. We define related concepts and the efficiency (Definition 2) of a gradient.

### 2.1 DEFINITIONS

We have training data $\mathcal{D}_{\text{train}}$ and test data $\mathcal{D}_{\text{test}}$, both non-empty. They are disjoint in a systematic generalization setting. Their union is all data $\mathcal{D}_{\text{all}}$, i.e., $\mathcal{D}_{\text{all}} = \mathcal{D}_{\text{train}} \dot{\cup} \mathcal{D}_{\text{test}}$. A loss for dataset $\mathcal{D}$ is the average loss of samples in the dataset.

$$\mathcal{L}(\mathcal{D}) = \frac{1}{|\mathcal{D}|} \sum_{(x,y) \in \mathcal{D}} \mathcal{L}(x, y) \in \mathbb{R}$$

Accordingly, we have three losses $\mathcal{L}_{\text{train}} = \mathcal{L}(\mathcal{D}_{\text{train}}), \mathcal{L}_{\text{test}} = \mathcal{L}(\mathcal{D}_{\text{test}}), \mathcal{L}_{\text{all}} = \mathcal{L}(\mathcal{D}_{\text{all}})$. Suppose the model has $n$ parameters $\theta \in \mathbb{R}^n$. We have three gradients $\nabla_\theta \mathcal{L}_{\text{train}}, \nabla_\theta \mathcal{L}_{\text{test}}, \nabla_\theta \mathcal{L}_{\text{all}} \in \mathbb{R}^n$. We use $\nabla_\theta \mathcal{L}_{\text{all}}$ as the alternative gradient to compare with the training gradient.

Since we study the gradient of a loss w.r.t. parameters, the input space is the model parameter space $\mathbb{R}^n$, and the output is the loss $\mathbb{R}$. We have a function $f : \mathbb{R}^n \to \mathbb{R}$, an input variable $\theta \in \mathbb{R}^n$, an input point $\theta_0 \in \mathbb{R}^n$, a vector in input space $\mathbf{u} \in \mathbb{R}^n$ and a scalar $h \in \mathbb{R}$. We use the directional derivative[1] (e.g., Strang (1991), p.490) to study the loss change when applying a gradient to update parameters.

---

[1]The definition sometimes requires $\mathbf{u}$ to be a unit vector. It is equivalent to using a unit vector $\hat{\mathbf{u}}$ here.

**Definition 1** (Directional derivative). *The directional derivative $D_{\mathbf{u}}f$ is the rate at which the function $f$ changes at a point $\theta_0$ in the direction $\mathbf{u}$. Suppose $\mathbf{u} \neq 0$.*

$$D_{\mathbf{u}}f(\theta_0) = \lim_{h \to 0} \frac{f(\theta_0 + h\hat{\mathbf{u}}) - f(\theta_0)}{h} = \nabla_\theta f \cdot \hat{\mathbf{u}} \in \mathbb{R}$$

*Provided the limit exists. $\hat{\mathbf{u}} = \frac{\mathbf{u}}{|\mathbf{u}|}$ is a unit vector.*

Directional derivative means the amount of output change when an input point is moved to a direction by a unit amount. We do not need the magnitude of the direction because the directional derivative is used in two cases. The magnitude is canceled when the ratio of two directional derivatives is computed. Only the sign (plus, minus, or zero) of the directional derivative is used.

We extend the definition for zero vector cases to cover zero gradients.

$$D(f, \mathbf{u}) = \begin{cases} D_{\mathbf{u}}f, & \text{if } \mathbf{u} \neq 0 \\ 0, & \text{if } \mathbf{u} = 0 \end{cases}$$

It has the same sign as $\nabla_\theta f \cdot \mathbf{u}$. We will frequently use whether the alternative gradient reduces training loss, so we name this threshold for convenience.

$$\Delta = D(\mathcal{L}_{\text{train}}, \nabla_\theta \mathcal{L}_{\text{all}}) \in \mathbb{R}$$

Note that a gradient is reversed by a negative sign when applied to update parameters. So the alternative gradient reduces training loss if $\Delta > 0$, increases it if $\Delta < 0$, and keeps it if $\Delta = 0$.

When a gradient updates parameters, both training and test losses change. We define the directional derivative ratio to find the efficiency by comparing two directional derivatives (Figure 2). Suppose we have another function $g : \mathbb{R}^n \to \mathbb{R}$.

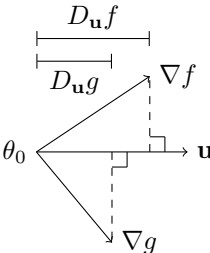

**Definition 2** (Directional derivative ratio, DDR). *The directional derivative ratio $DDR(f, g, \mathbf{u})$ of two functions $f, g$ and a direction $\mathbf{u}$ is the ratio of their directional derivatives at a point $\theta_0$ in the direction. Suppose $\mathbf{u} \neq 0$ and $D_{\mathbf{u}}g(\theta_0) \neq 0$.*

$$DDR(f, g, \mathbf{u}) = \frac{D_{\mathbf{u}}f(\theta_0)}{D_{\mathbf{u}}g(\theta_0)} = \frac{\nabla_\theta f(\theta_0) \cdot \mathbf{u}}{\nabla_\theta g(\theta_0) \cdot \mathbf{u}} \in \mathbb{R}$$

If $f$ is test loss, $g$ is training loss, and $\mathbf{u}$ is a gradient, DDR means the ratio of the changes in test and training losses when a gradient is applied to update parameters. It is also the amount of test loss change per unit training loss change. So it is the efficiency of the gradient to reduce the test loss with training loss reduction.

Figure 2: Illustration in parameter space. $f, g$ are loss functions. $\mathbf{u}$ is a direction. $DDR(f, g, \mathbf{u})$ is the ratio of $D_{\mathbf{u}}f$ and $D_{\mathbf{u}}g$.

We compare a training gradient and an alternative gradient at each training step. So we define criteria (Definition 3,4) to tell whether a gradient is better than another. Please also refer to Figure 1 for intuition. For simplicity, we call a case biased[2] if the alternative gradient is better than the training gradient. It is unbiased if both gradients are equally good. We will prove in Lemma 1 that the definitions cover, or partition, all possible cases. Note that (3), (4), and (5) are numerically rare cases because they require equal signs to hold.

**Definition 3** (Biased cases). *The following cases are biased.*

*(1) The alternative gradient reduces training loss, and it reduces more test loss per unit training loss reduction than the training gradient does.*

$$\Delta > 0 \quad and \quad DDR(\mathcal{L}_{test}, \mathcal{L}_{train}, \nabla_\theta \mathcal{L}_{train}) < DDR(\mathcal{L}_{test}, \mathcal{L}_{train}, \nabla_\theta \mathcal{L}_{all})$$

*(2) The alternative gradient increases training loss and reduces test loss, while the training gradient increases test loss.*

$$\Delta < 0 \quad and \quad D(\mathcal{L}_{test}, \nabla_\theta \mathcal{L}_{train}) < 0 < D(\mathcal{L}_{test}, \nabla_\theta \mathcal{L}_{all})$$

---

[2]We use the word "bias" because the training gradient can be regarded as a biased or unbiased estimator of the alternative gradient.

*(3) The alternative gradient keeps training loss and reduces/keeps test loss, while the training gradient increases/keeps test loss. The two gradients do not keep test loss simultaneously.*

$$\Delta = 0 \quad and \quad D(\mathcal{L}_{test}, \nabla_\theta \mathcal{L}_{train}) \leq 0 \leq D(\mathcal{L}_{test}, \nabla_\theta \mathcal{L}_{all}), \textit{ but not both equal}$$

**Definition 4** (Unbiased cases). *The following cases are unbiased.*

*(4) The alternative gradient reduces training loss, and it reduces the same amount of test loss per unit training loss reduction as the training gradient does.*

$$\Delta > 0 \quad and \quad DDR(\mathcal{L}_{test}, \mathcal{L}_{train}, \nabla_\theta \mathcal{L}_{train}) = DDR(\mathcal{L}_{test}, \mathcal{L}_{train}, \nabla_\theta \mathcal{L}_{all})$$

*(5) The alternative gradient keeps training loss, and both gradients keep test loss.*

$$\Delta = 0 \quad and \quad D(\mathcal{L}_{test}, \nabla_\theta \mathcal{L}_{train}) = D(\mathcal{L}_{test}, \nabla_\theta \mathcal{L}_{all}) = 0$$

For two reasons, DDR is not used for $\Delta \leq 0$. First, we can define whether there is a bias without DDR. Second, it is not straightforward to translate DDR as efficiency in such cases. It is because we study the efficiency while reducing training loss, but the alternative gradient increases or keeps training loss here. We will discuss more when defining Unified DDR (Definition 5).

## 2.2 DERIVATIONS

We compare the two gradients. We first derive propositions for different $\Delta$ values. We then confirm that the definition of biases is valid and derive a theorem. The proofs are in Appendix A.

**Proposition 1** (Reduce training loss). *If $\Delta > 0$,*

$$DDR(\mathcal{L}_{test}, \mathcal{L}_{train}, \nabla_\theta \mathcal{L}_{train}) \leq DDR(\mathcal{L}_{test}, \mathcal{L}_{train}, \nabla_\theta \mathcal{L}_{all})$$

*The equal sign holds if and only if $\nabla_\theta \mathcal{L}_{test} = 0$ or $\sin(\nabla_\theta \mathcal{L}_{train}, \nabla_\theta \mathcal{L}_{test}) = 0$.*

This proposition corresponds to the definitions (1) and (4). It tells that the two definitions cover all cases for $\Delta > 0$ because the training gradient is at most equally efficient to the alternative gradient. It also provides the conditions for equal efficiency.

**Proposition 2** (Increase training loss). *If $\Delta < 0$,*

$$D(\mathcal{L}_{test}, \nabla_\theta \mathcal{L}_{train}) < 0 < D(\mathcal{L}_{test}, \nabla_\theta \mathcal{L}_{all})$$

This proposition corresponds to definition (2). It tells that the bias always exists when $\Delta < 0$.

**Proposition 3** (Keep training loss). *If $\Delta = 0$,*

$$D(\mathcal{L}_{test}, \nabla_\theta \mathcal{L}_{train}) \leq 0 \leq D(\mathcal{L}_{test}, \nabla_\theta \mathcal{L}_{all})$$

*The two equal signs hold simultaneously if and only if $\nabla_\theta \mathcal{L}_{test} = 0$.*

This proposition corresponds to definitions (3) and (5). The condition $\Delta = 0$ is rare to hold because it requires an equation to hold. It also tells when the bias is avoided in such cases.

The propositions imply that there are constraints for possible cases. We consider all biased cases as a set and all unbiased cases as another set. Then Lemma 1 says the two sets partition the set of all available cases. The union of the two sets equals the set of all available cases. The two sets are disjoint and both non-empty.

**Lemma 1** (Partition). *The set of biased cases and the set of unbiased cases are a partition of the set of all available cases.*

We then draw a theorem for the bias. i.e., an alternative gradient is better than a training one.

**Theorem 1** (Bias). *There is a bias if and only if neither of the following holds.*

$$(A) \quad \nabla_\theta \mathcal{L}_{train} = \nabla_\theta \mathcal{L}_{all} = 0 \qquad\qquad (B) \quad \cos(\nabla_\theta \mathcal{L}_{train}, \nabla_\theta \mathcal{L}_{all}) = 1$$

It means training and alternative gradients are either both zero or point in the same direction to avoid bias. Both (A) and (B) contain equal signs, which are generally difficult to hold. Please refer to Section 4 for more discussions.

## 3 EXPERIMENTS

We run experiments to verify the derivations and observe differences between the training and the alternative gradients. We cover different standard deep neural network models. The details of networks and experiments can be found in Appendix B.

### 3.1 METRIC

In the previous section, we use DDR only for $\Delta > 0$, and still do not have a quantitative metric for other cases. So we extend DDR to all cases by defining Unified DDR (UDDR).

**Definition 5** (UDDR). *If $\nabla_\theta g(\theta_0) \cdot \mathbf{u} \neq 0$,*

$$UDDR(f, g, \mathbf{u}) = \frac{D_\mathbf{u} f(\theta_0)}{|D_\mathbf{u} g(\theta_0)|} = \frac{\nabla_\theta f(\theta_0) \cdot \mathbf{u}}{|\nabla_\theta g(\theta_0) \cdot \mathbf{u}|} \in \mathbb{R}$$

*If $\nabla_\theta g(\theta_0) \cdot \mathbf{u} = 0$, $UDDR(f, g, \mathbf{u}) = +\infty/0/-\infty$, for $D_\mathbf{u} f(\theta_0) > / = / < 0$, respectively.*

It means the change of $f$ when $g$ moves to its original direction by a unit amount. When $\Delta > 0$, UDDR equals DDR. UDDR maintains the inequality property.

**Corollary 1** (UDDR inequality).

$$UDDR(\mathcal{L}_{test}, \mathcal{L}_{train}, \nabla_\theta \mathcal{L}_{train}) \leq UDDR(\mathcal{L}_{test}, \mathcal{L}_{train}, \nabla_\theta \mathcal{L}_{all})$$

*The equal sign holds if and only if a case is unbiased.*

An unbiased case means (A) or (B) in Theorem 1 holds. We also derive the difference in UDDR values between the alternative and the training gradients (Section 4.1).

### 3.2 DATA

We use image and text datasets. Image datasets include human faces (CelebA) and natural scenes (NICO++). Text datasets have sequence prediction (CFQ) and classification (Amazon reviews). We also has experiments with disentangled input data (Appendix B.3), which corresponds to oracle pre-training of representation learning.

**Multi-class classification** We run experiments for multi-class classification tasks with two network outputs. For image data, we use the NICO++ dataset (Zhang et al., 2022). We use five foregrounds as the first output label and five backgrounds as the second. For text data, we use Amazon reviews (Ni et al., 2019). We use five categories as the first output label and five ratings as the second. We use the Fashion dataset (Xiao et al., 2017) with ten classes for fully connected networks and render them with ten colors as the second label. Please refer to Figure 3 and Table 1 for examples.

We design training and test label combinations for systematic generalization. A combination with output $y_1$ and $y_2$ is a training one, if $y_2$ is one of $k$ classes $\{y_1, y_1 + 1, \ldots, y_1 + k - 1\}$ (we use modular for the class labels). $k$ is five if there are ten classes and three if there are five classes. The test label combinations are the remaining ones. Training and test label combinations are mutually exclusive, but test labels for each output factor are seen in training.

**CelebA and CFQ** We also run experiments on the CelebA dataset (Liu et al., 2015) and the CFQ dataset (Keysers et al., 2020). CelebA is used for fully connected, convolutional, residual networks and Vision Transformer. CFQ dataset is used for LSTM and Transformer.

CelebA dataset has face images of various celebrities. We follow the setup from previous work (Sagawa et al., 2020; Yao et al., 2022) and use two attributes as factors. The first factor is hair color "blond" or "non-blond". The second factor is male or female. The test data is "blond hair male," and the training data is the other three combinations.

Compositional Freebase Questions (CFQ) is a large realistic semantic parsing dataset for systematic generalization. The inputs are natural language questions (e.g., Who directed Elysium), and the outputs are SPARQL queries for the Freebase knowledge base. Samples are generated from rules. "Maximum compound divergence" (MCD) splits maximize the rule combination divergence while keeping a small rule divergence between train and test sets. The results for the first MCD split are in Figure 5, and the others are in Appendix B.

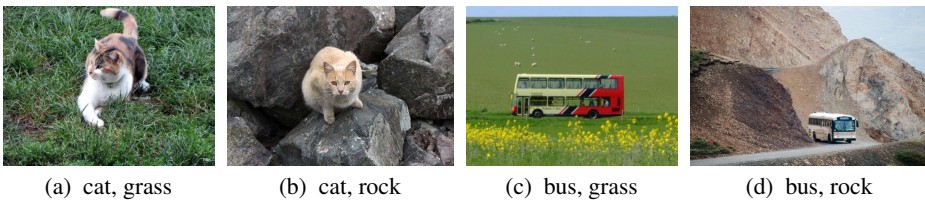

| (a) cat, grass | (b) cat, rock | (c) bus, grass | (d) bus, rock |

Figure 3: Examples of image data.

| Category | Rating | Review |
|---|---|---|
| Book | 5 | quick read from the most excellent author. fun |
| Book | 1 | It had tears, some label covering a defect and also wrinkled pages. |
| Electronics | 5 | Works perfectly, better than the original cable. |
| Electronics | 1 | DID NOT fit as described to accommodate the TV size! |

Table 1: Examples of text data.

## 3.3 RESULTS

**Fully Connected Network**: We use a five-layer fully connected neural network with a flattened image input.

**Convolutional Network**: We use a convolutional neural network with three convolutional layers and two fully connected layers.

**Residual Network**: We use ResNet50 (He et al., 2016).

**Vision Transformer**: We use Vision Transformer (Dosovitskiy et al., 2021) with one fully connected layer for each patch, three attention layers, and one fully connected layer.

**LSTM**: For classification problems, we use stacked LSTM models with an embedding layer, three bidirectional LSTM layers, and one fully connected layer. For sequence prediction problems, we use an embedding layer, a single-layer bidirectional LSTM as an encoder, and a single-layer LSTM with attention as a decoder.

**Transformer**: We use Transformer (Vaswani et al., 2017). For classification problems, we only use the encoder. It has one embedding layer, three hidden layers, and one fully connected layer. For sequence prediction problems, we use one embedding layer and four hidden layers.

**Summary of results**  Figure 4 and Figure 5 contain the results[3]. We repeat each experiment with different random seeds five times and plot the means and the standard deviations. It shows that, for each evaluation, the UDDR value is higher for the alternative gradient than the training gradient. It verifies the derivations with different architectures. It also shows significant differences between the scores in general, which indicates that bias is common in deep learning.

## 4 DISCUSSIONS

We discuss more details of the bias, and derive the difference in the UDDR between the alternative and the training gradients. We also propose a requirement on models for systematic generalization. Please also refer to variants of gradients in Appendix C.

### 4.1 UDDR GAP

We have the expression of the UDDR gap between the alternative and training gradient. When $\Delta > 0$, it is the expression of the DDR gap.

---

[3]We smooth the values for visualization. We compute the average for every 50 iterations. We then convert with function $f(x) = \text{sign}(x)[\ln(|x| + 1)]$, which log scales and is monotonic.

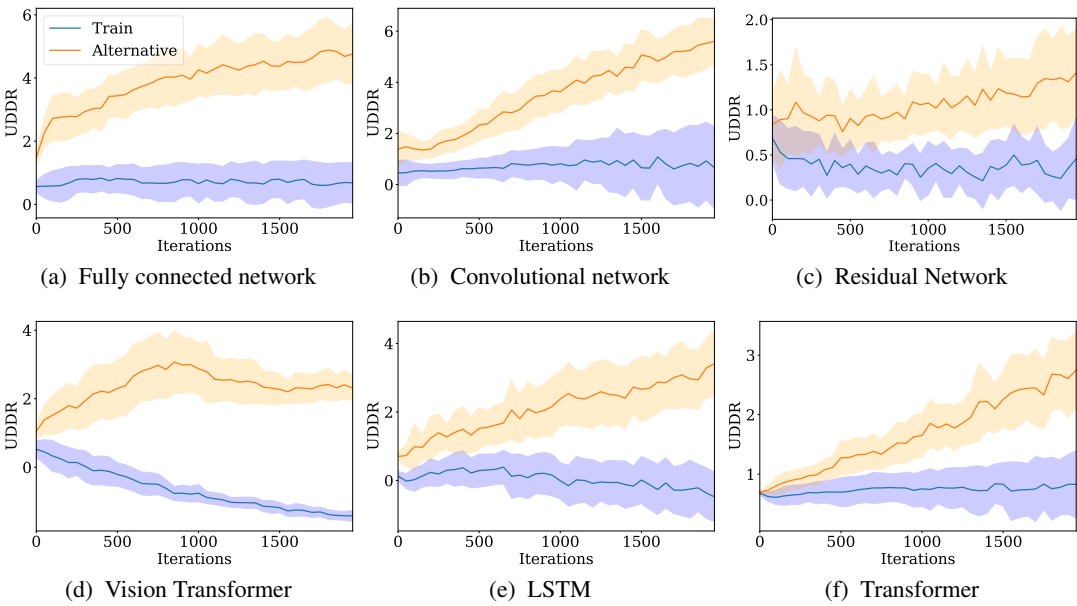

Figure 4: UDDR during training for multi-class classification datasets. We plot in log scale while keeping signs. In general, the score is significantly larger for alternative gradients than for training gradients in each experiment. It indicates that bias is common in deep learning training.

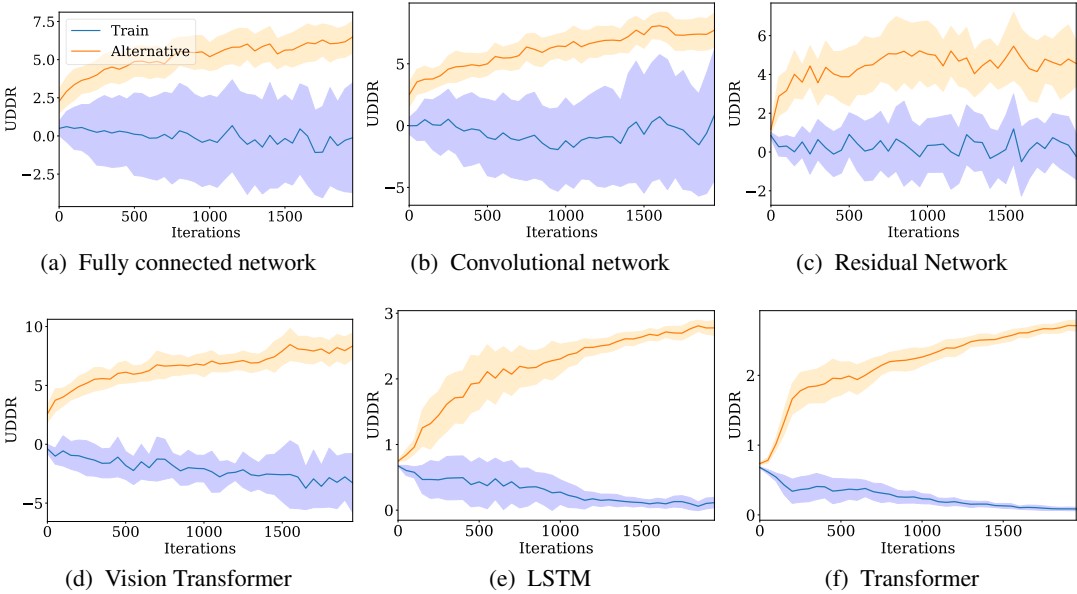

Figure 5: UDDR during training for CelebA dataset (Liu et al., 2015) and CFQ dataset, MCD split (Keysers et al., 2020).

**Lemma 2** (UDDR gap).

$$\Delta UDDR = UDDR(\mathcal{L}_{test}, \mathcal{L}_{train}, \nabla_\theta \mathcal{L}_{all}) - UDDR(\mathcal{L}_{test}, \mathcal{L}_{train}, \nabla_\theta \mathcal{L}_{train})$$

$$= \begin{cases} 0, & \text{if } \nabla_\theta \mathcal{L}_{test} = 0, \text{ otherwise} \\ \frac{|b|^2 \sin^2(a,b)}{(a+b)^T a}, & \text{if } \Delta > 0 \\ \frac{|b|^2 \sin^2(a,b) - 2(a+b)^T b}{(a+b)^T a}, & \text{if } \Delta < 0 \\ +\infty, & \text{if } \Delta = 0 \end{cases}$$

*Where $a = |\mathcal{D}_{train}| \nabla_\theta \mathcal{L}_{train}$ and $b = |\mathcal{D}_{test}| \nabla_\theta \mathcal{L}_{test}$.*

### 4.2 A REQUIREMENT FOR SYSTEMATIC GENERALIZATION

We propose a requirement for systematic generalization. Based on it, we discuss why standard models do not generalize, and analyze methods that meet the requirement.

A neural network can be decomposed into sub-networks. If a sub-network has unseen test input, the input corresponds to a new factor combination, so the sub-network still has a systematic generalization problem. So we propose the following requirements. Note that it is a necessary requirement, so it alone may not achieve the generalization.

**Requirement** *Systematic generalization requires that a model can be decomposed into sub-networks, each with seen test inputs.*

**Why standard models do not achieve systematic generalization** We consider standard models without special design of architecture or regularization. For example, in a fully connected network, a node in the first hidden layer is the output of a sub-network whose input is the model input. Since this sub-network is not further decomposable and has unseen test inputs, it does not meet the requirement. In such cases, the trained models cannot be decomposed into sub-networks with seen test inputs. So, these designs are not likely to achieve systematic generalization. Also, the experiments in Sections 3 show that the training and alternative gradients have different UDDR values, so they do not point in the same direction.

**Methods that meet the requirements** With neural networks, the approaches to systematic generalization include hybrid approaches and connectionist approaches.

Hybrid methods (Liu et al., 2020; Chen et al., 2020) combine symbol processing and neural networks. Symbols keep the (pre-defined) representations across training and test data, so the sub-network has seen test inputs.

Connectionist approaches encourage neural networks to map different representations to the same one. For example, reduce representation entropy (Li et al., 2019) or use discrete representations (Liu et al., 2021). Also, attention mechanism reduces input size, making it easier to have a seen test input.

### 4.3 I.I.D. CASES

The theorem also applies to i.i.d. generalization problems as a special case. In i.i.d. settings, both training and test data are independently drawn from the identical distribution. It means that the expectations of the training and the alternative gradients are the same. So, the gradients are either both zero or point in the same direction. By Theorem 1, the bias is avoided.

## 5 RELATED WORK

**Systematic generalization and deep learning** Systematic generalization (Fodor & Pylyshyn, 1988; Lake & Baroni, 2018; Bahdanau et al., 2019), or compositional generalization, is considered the "Great Move" of evolution, caused by the need to process an increasing amount and diversity of environmental information (Newell, 1990). Cognitive scientists see it as central for an organism to view the world (Gallistel & King, 2011). Studies indicate it is related to the prefrontal cortex (Robin & Holyoak, 1995). It was discussed that commonsense is critical (Mccarthy, 1959; Lenat et al., 1986)

for systematic generalization, and recent works aim to find general prior knowledge (Goyal & Bengio, 2020), e.g., Consciousness Prior (Bengio, 2017). Levels of systematicity were defined (Hadley, 1992; Niklasson & van Gelder, 1994), and types of tests were summarized (Hupkes et al., 2020). We focus on the primary case with an unseen combination of seen factor values.

A closely related field is causal learning, rooted in the eighteenth-century (Hume, 2003) and classical fields of AI (Pearl, 2003). It was mainly explored from statistical perspectives (Pearl, 2009; Peters et al., 2016; Greenland et al., 1999; Pearl, 2018) with do-calculus (Pearl, 1995; 2009) and interventions (Peters et al., 2016). The causation forms Independent Causal Mechanisms (ICMs) (Peters et al., 2017; Schölkopf et al., 2021). Systematic generalization is the counterfactual when the joint input distribution is intervened to have new values with zero probability in training (covariate shift). This work indicates that standard neural networks do not prefer to learn ICMs.

Parallel Distributed Processing (PDP) models (Rumelhart et al., 1986) use Connectionist models with distributed representations, which describe an object in terms of a set of factors. Though they have the potential to combine the factors to create unseen object representations (Hinton, 1990), it was criticized that they do not address systematic generalization in general (Fodor & Pylyshyn, 1988; Marcus, 1998). Deep learning is a recent PDP model with many achievements (LeCun et al., 2015; He et al., 2016). The improvements in i.i.d. problems encourage to equip deep learning with systematic generalization.

**Recent directions** In addition to architecture design (Russin et al., 2019; Andreas et al., 2016) and data augmentation (Andreas, 2020; Akyürek et al., 2021; Jia & Liang, 2016), the main perspectives for systematic generalization approaches include disentangled representation learning, attention mechanism, and meta-learning.

Disentangled representation (Bengio et al., 2013) is learned in unsupervised manners. Early methods learn the representation from statistical independence (Higgins et al., 2017; Locatello et al., 2019). Later, the definition of disentangled representation was proposed with symmetry transformation (Higgins et al., 2018). It leads to Symmetry-based Disentangled Representation Learning (Caselles-Dupré et al., 2019; Painter et al., 2020; Pfau et al., 2020). A disentangled representation learning model can be used as a feature extractor for other systematic generalization tasks.

Attention mechanisms are widely used in neural networks (Bahdanau et al., 2015). Transformers (Vaswani et al., 2017) are modern neural network architectures with self-attention. Recurrent Independent Mechanisms (Goyal et al., 2021b) use attention and the name of the incoming nodes for variable binding. Global workspace (Goyal et al., 2021a) improves them by using limited-capacity global communication to enable the exchangeability of knowledge. Discrete-valued communication bottleneck (Liu et al., 2021) further enhances systematic generalization ability.

Meta-learning (Lake, 2019) usually designs a series of training tasks for learning a meta-learner and uses it in a target task. Each task has training and test data, where test data requires systematic generalization from training data. When ICMs are available, they can be used to generate meta-learning tasks (Schölkopf et al., 2021). Meta-reinforcement learning was used for causal reasoning (Dasgupta et al., 2019). Meta-learning can also capture the adaptation speed to discover causal relations (Bengio et al., 2020; Ke et al., 2019).

Deep learning is a fast-growing field, and many efforts focus on designing architectures and algorithms to improve its performance. This paper studies from a gradient perspective, by looking at the relation between gradient directions and the generalization.

## 6 CONCLUSION

This paper investigates the relation between gradient directions and systematic generalization. We propose a formulation to treat training loss as a resource and define DDR to measure the efficiency of consuming it. We derive that there is a bias in generalization if the training and the alternative gradients have different directions. We show the bias in various standard deep neural networks. We also discuss a requirement for the generalization. We hope this study provides a new understanding of systematic generalization mechanisms in deep learning and helps to improve machine learning algorithms for a higher level of artificial intelligence.

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

## A  PROOFS

Symbols have the following correspondence in the proofs.

$$a = |\mathcal{D}_{\text{train}}|\nabla_\theta \mathcal{L}_{\text{train}} \qquad\qquad b = |\mathcal{D}_{\text{test}}|\nabla_\theta \mathcal{L}_{\text{test}}$$

### A.1  PROPOSITION 1

**Lemma 3** (Difference). $\forall a, b \in \mathbb{R}^n$, if $(a+b)^T a \neq 0$, then the following equation holds.

$$\frac{(a+b)^T b}{(a+b)^T a} - \frac{a^T b}{a^T a} = \begin{cases} \frac{|b|^2 \sin^2(a,b)}{(a+b)^T a}, & \text{If } b \neq 0 \\ 0, & \text{If } b = 0 \end{cases}$$

*Proof.*

$$(a+b)^T a \neq 0 \implies a \neq 0 \implies a^T a \neq 0$$

So $\frac{a^T b}{a^T a}$ is well-defined. If $b = 0$, the result is 0. If $b \neq 0$,

$$\begin{aligned}
\frac{(a+b)^T b}{(a+b)^T a} - \frac{a^T b}{a^T a} &= \frac{(a+b)^T b a^T a - a^T b (a+b)^T a}{(a+b)^T a a^T a} \\
&= \frac{a^T b a^T a + b^T b a^T a - a^T b a^T a - a^T b b^T a}{(a+b)^T a a^T a} \\
&= \frac{b^T b a^T a - a^T b b^T a}{(a+b)^T a a^T a} \\
&= \frac{|b|^2 |a|^2 - |a||b|\cos(a,b)|a||b|\cos(a,b)}{|a|^2 (a+b)^T a} \\
&= \frac{|a|^2 |b|^2 (1 - \cos^2(a,b))}{|a|^2 (a+b)^T a} \\
&= \frac{|b|^2 \sin^2(a,b)}{(a+b)^T a}
\end{aligned}$$

$\square$

**Lemma 4** (Inequality). $\forall a, b \in \mathbb{R}^n$, if $(a+b)^T a > 0$, then the following inequation holds.

$$\frac{a^T b}{a^T a} \leq \frac{(a+b)^T b}{(a+b)^T a}$$

*The equal sign holds if and only if $b = 0$ or $\sin(a,b) = 0$.*

*Proof.* We use Lemma 3 and suppose $(a+b)^T a > 0$. If $b \neq 0$, we have

$$\frac{(a+b)^T b}{(a+b)^T a} - \frac{a^T b}{a^T a} = \frac{|b|^2 \sin^2(a,b)}{(a+b)^T a} \geq 0$$

The equal sign holds if and only if $\sin(a,b) = 0$. If $b = 0$, $\frac{(a+b)^T b}{(a+b)^T a} - \frac{a^T b}{a^T a} = 0$.

Therefore, $\frac{a^T b}{a^T a} \leq \frac{(a+b)^T b}{(a+b)^T a}$, and the equal sign holds if and only if $b = 0$ or $\sin(a,b) = 0$.

$\square$

**Lemma 5** (Gradient relations).

$$|\mathcal{D}_{all}|\nabla_\theta \mathcal{L}_{all} = |\mathcal{D}_{train}|\nabla_\theta \mathcal{L}_{train} + |\mathcal{D}_{test}|\nabla_\theta \mathcal{L}_{test}$$

*Proof.* By definition, we have $\mathcal{D}_{\text{all}} = \mathcal{D}_{\text{train}} \dot{\cup} \mathcal{D}_{\text{test}}$.

$$\implies \sum_{(x,y)\in\mathcal{D}_{\text{all}}} \mathcal{L}(x,y) = \sum_{(x,y)\in\mathcal{D}_{\text{train}}} \mathcal{L}(x,y) + \sum_{(x,y)\in\mathcal{D}_{\text{test}}} \mathcal{L}(x,y)$$

$$\implies |\mathcal{D}_{\text{all}}|\mathcal{L}_{\text{all}} = |\mathcal{D}_{\text{train}}|\mathcal{L}_{\text{train}} + |\mathcal{D}_{\text{test}}|\mathcal{L}_{\text{test}}$$

$$\implies |\mathcal{D}_{\text{all}}|\nabla_\theta\mathcal{L}_{\text{all}} = |\mathcal{D}_{\text{train}}|\nabla_\theta\mathcal{L}_{\text{train}} + |\mathcal{D}_{\text{test}}|\nabla_\theta\mathcal{L}_{\text{test}}$$

$\square$

**Lemma 6** (Coefficients)**.**

$$\forall \alpha, \beta, \gamma \in \mathbb{R}, \beta \neq 0, \gamma \neq 0 : DDR(\alpha a, \beta b, \gamma c) = \frac{\alpha}{\beta} DDR(a, b, c)$$

*Proof.*

$$\text{DDR}(\alpha a, \beta b, \gamma c) = \frac{(\alpha\nabla a)^T(\gamma c)}{(\beta\nabla b)^T(\gamma c)} = \frac{\alpha}{\beta}\frac{(\nabla a)^T c}{(\nabla b)^T c} = \frac{\alpha}{\beta}\text{DDR}(a, b, c)$$

$\square$

**Proposition 1** (Reduce training loss)**.** *If $\Delta > 0$,*

$$DDR(\mathcal{L}_{\text{test}}, \mathcal{L}_{\text{train}}, \nabla_\theta\mathcal{L}_{\text{train}}) \leq DDR(\mathcal{L}_{\text{test}}, \mathcal{L}_{\text{train}}, \nabla_\theta\mathcal{L}_{\text{all}})$$

*The equal sign holds if and only if $\nabla_\theta\mathcal{L}_{\text{test}} = 0$ or $\sin(\nabla_\theta\mathcal{L}_{\text{train}}, \nabla_\theta\mathcal{L}_{\text{test}}) = 0$.*

*Proof.* This proposition is equivalent to the following inequation because of Lemma 6.

$$\text{DDR}(|\mathcal{D}_{\text{test}}|\mathcal{L}_{\text{test}}, |\mathcal{D}_{\text{train}}|\mathcal{L}_{\text{train}}, |\mathcal{D}_{\text{train}}|\nabla_\theta\mathcal{L}_{\text{train}}) \leq \text{DDR}(|\mathcal{D}_{\text{test}}|\mathcal{L}_{\text{test}}, |\mathcal{D}_{\text{train}}|\mathcal{L}_{\text{train}}, |\mathcal{D}_{\text{all}}|\nabla_\theta\mathcal{L}_{\text{all}})$$

With Lemma 5, Lemma 4 applies. $\square$

## A.2 PROPOSITION 2

**Lemma 7** (Inequality)**.**

$$\forall a, b \in \mathbb{R}^n : \quad (a+b)^T a < 0 \implies a^T b < 0 < (a+b)^T b$$

*Proof.* Given $(a+b)^T a < 0$,

$$a^T b \leq a^T a + b^T a = (a+b)^T a < 0$$
$$(a+b)^T b > (a+b)^T b + (a+b)^T a = (a+b)^T(a+b) \geq 0$$

Therefore, $a^T b < 0 < (a+b)^T b$. $\square$

**Proposition 2** (Increase training loss)**.** *If $\Delta < 0$,*

$$D(\mathcal{L}_{\text{test}}, \nabla_\theta\mathcal{L}_{\text{train}}) < 0 < D(\mathcal{L}_{\text{test}}, \nabla_\theta\mathcal{L}_{\text{all}})$$

*Proof.* With Lemma 5 and Lemma 7, we have

$$|\mathcal{D}_{\text{test}}|\nabla_\theta\mathcal{L}_{\text{test}} \cdot |\mathcal{D}_{\text{train}}|\nabla_\theta\mathcal{L}_{\text{train}} < 0 < |\mathcal{D}_{\text{test}}|\nabla_\theta\mathcal{L}_{\text{test}} \cdot |\mathcal{D}_{\text{all}}|\nabla_\theta\mathcal{L}_{\text{all}}$$

The sample numbers are all positive, so they do not change the comparison with zero. Therefore, $\nabla_\theta\mathcal{L}_{\text{test}} \cdot \nabla_\theta\mathcal{L}_{\text{train}} < 0 < \nabla_\theta\mathcal{L}_{\text{test}} \cdot \nabla_\theta\mathcal{L}_{\text{all}}$. $\square$

### A.3 PROPOSITION 3

**Lemma 8** (Equality).

$$\forall a, b \in \mathbb{R}^n: \quad (a+b)^T a = 0 \implies a^T b \leq 0 \leq (a+b)^T b$$

*The two equal signs hold simultaneously if and only if $b = 0$.*

*Proof.* Given $(a+b)^T a = 0$,

$$a^T b \leq a^T a + b^T a = (a+b)^T a = 0$$
$$(a+b)^T b = (a+b)^T b + (a+b)^T a = (a+b)^T(a+b) \geq 0$$

Therefore, $a^T b \leq 0 \leq (a+b)^T b$.

To prove equality,

$$b = 0 \implies a^T b = 0 = (a+b)^T b$$
$$a^T b = 0 = (a+b)^T b \implies b^T b = 0 \implies b = 0$$

So the two equal signs hold simultaneously if and only if $b = 0$. $\qquad\square$

**Proposition 3** (Keep training loss). *If $\Delta = 0$,*

$$D(\mathcal{L}_{test}, \nabla_\theta \mathcal{L}_{train}) \leq 0 \leq D(\mathcal{L}_{test}, \nabla_\theta \mathcal{L}_{all})$$

*The two equal signs hold simultaneously if and only if $\nabla_\theta \mathcal{L}_{test} = 0$.*

*Proof.* With Lemma 5 and Lemma 8, we have

$$|\mathcal{D}_{\text{test}}|\nabla_\theta \mathcal{L}_{\text{test}} \cdot |\mathcal{D}_{\text{train}}|\nabla_\theta \mathcal{L}_{\text{train}} \leq 0 \leq |\mathcal{D}_{\text{test}}|\nabla_\theta \mathcal{L}_{\text{test}} \cdot |\mathcal{D}_{\text{all}}|\nabla_\theta \mathcal{L}_{\text{all}}$$

The two equal signs hold simultaneously if and only if $|\mathcal{D}_{\text{test}}|\mathcal{L}_{\text{test}} = 0$. The sample numbers are all positive, so they do not change the comparison with zero.

Therefore, $\nabla_\theta \mathcal{L}_{\text{test}} \cdot \nabla_\theta \mathcal{L}_{\text{train}} \leq 0 \leq \nabla_\theta \mathcal{L}_{\text{test}} \cdot \nabla_\theta \mathcal{L}_{\text{all}}$. The two equal signs hold simultaneously if and only if $\nabla_\theta \mathcal{L}_{\text{test}} = 0$. $\qquad\square$

### A.4 LEMMA 1

**Lemma 1** (Partition). *The set of biased cases and the set of unbiased cases are a partition of the set of all available cases.*

*Proof.* We prove the union has a full cover, the intersection is disjoint, and the two sets are non-empty. (1) (2) (3) are from Definition 3, and (4) (5) are from Definition 4.

(a) Union has a full cover
There are three disjoint cases, and we look into each of them.
If $\Delta > 0$, by Proposition 1, (1) and (4) cover all the cases.
If $\Delta < 0$, by Proposition 2, (2) covers all the cases.
If $\Delta = 0$, by Proposition 3, (3) and (5) covers all the case.
Therefore, all the cases are covered by the definitions.

(b) Intersection is disjoint
If $\Delta > 0$, (1) and (4) are disjoint.
If $\Delta < 0$, there are only biased cases (2), so the intersection is empty.
If $\Delta = 0$, (3) and (5) are disjoint.
Therefore, biased and unbiased cases do not overlap.

(c) Non-empty
Biased case: we consider a case of $\nabla_\theta \mathcal{L}_{\text{test}} = -\frac{2|\mathcal{D}_{\text{train}}|}{|\mathcal{D}_{\text{test}}|} \nabla_\theta \mathcal{L}_{\text{train}} \neq 0$.

$$\nabla_\theta \mathcal{L}_{\text{train}} \cdot \nabla_\theta \mathcal{L}_{\text{all}} = \nabla_\theta \mathcal{L}_{\text{train}} \cdot \frac{1}{|\mathcal{D}_{\text{all}}|}(|\mathcal{D}_{\text{train}}|\nabla_\theta \mathcal{L}_{\text{train}} + |\mathcal{D}_{\text{test}}|\nabla_\theta \mathcal{L}_{\text{test}}) = -\frac{|\mathcal{D}_{\text{train}}|}{|\mathcal{D}_{\text{all}}|}|\nabla_\theta \mathcal{L}_{\text{train}}|^2 < 0$$

It follows that $\Delta < 0$. By Definition 3 (2) and Proposition 2, it is a biased case.
Unbiased case: $\nabla_\theta \mathcal{L}_{\text{test}} = 0$ is an unbiased case because of Definition 3 (5) and Proposition 3. $\quad\square$

## A.5 THEOREM 1

**Lemma 9** (Equivalence when reducing training loss)**.**

$$\Delta > 0 \text{ and } [\nabla_\theta \mathcal{L}_{test} = 0 \text{ or } \sin(\nabla_\theta \mathcal{L}_{train}, \nabla_\theta \mathcal{L}_{test}) = 0] \iff \cos(\nabla_\theta \mathcal{L}_{train}, \nabla_\theta \mathcal{L}_{all}) = 1$$

*Proof.* To prove " $\implies$ ",

$$\Delta > 0 \implies \nabla_\theta \mathcal{L}_{\text{train}} \cdot \nabla_\theta \mathcal{L}_{\text{all}} > 0 \implies \cos(\nabla_\theta \mathcal{L}_{\text{train}}, \nabla_\theta \mathcal{L}_{\text{all}}) > 0$$

If $\nabla_\theta \mathcal{L}_{\text{test}} = 0$,

$$\nabla_\theta \mathcal{L}_{\text{all}} = \frac{|\mathcal{D}_{\text{train}}|}{|\mathcal{D}_{\text{all}}|} \nabla_\theta \mathcal{L}_{\text{train}} + \frac{|\mathcal{D}_{\text{test}}|}{|\mathcal{D}_{\text{all}}|} \nabla_\theta \mathcal{L}_{\text{test}} = \frac{|\mathcal{D}_{\text{train}}|}{|\mathcal{D}_{\text{all}}|} \nabla_\theta \mathcal{L}_{\text{train}} \implies \cos(\nabla_\theta \mathcal{L}_{\text{train}}, \nabla_\theta \mathcal{L}_{\text{all}}) = 1$$

If $\sin(\nabla_\theta \mathcal{L}_{\text{train}}, \nabla_\theta \mathcal{L}_{\text{test}}) = 0$, then $\cos(\nabla_\theta \mathcal{L}_{\text{train}}, \nabla_\theta \mathcal{L}_{\text{all}}) = 1$.

To prove " $\impliedby$ ",

$$\cos(\nabla_\theta \mathcal{L}_{\text{train}}, \nabla_\theta \mathcal{L}_{\text{all}}) = 1 > 0 \implies \nabla_\theta \mathcal{L}_{\text{train}} \cdot \nabla_\theta \mathcal{L}_{\text{all}} > 0 \implies \Delta > 0$$

$$\cos(\nabla_\theta \mathcal{L}_{\text{train}}, \nabla_\theta \mathcal{L}_{\text{all}}) = 1 \implies \exists \alpha > 0 : \nabla_\theta \mathcal{L}_{\text{all}} = \alpha \nabla_\theta \mathcal{L}_{\text{train}}$$

$$\implies \nabla_\theta \mathcal{L}_{\text{test}} = \frac{|\mathcal{D}_{\text{all}}|}{|\mathcal{D}_{\text{test}}|} \nabla_\theta \mathcal{L}_{\text{all}} - \frac{|\mathcal{D}_{\text{train}}|}{|\mathcal{D}_{\text{test}}|} \nabla_\theta \mathcal{L}_{\text{train}} = \frac{\alpha |\mathcal{D}_{\text{all}}| - |\mathcal{D}_{\text{train}}|}{|\mathcal{D}_{\text{test}}|} \nabla_\theta \mathcal{L}_{\text{train}}$$

If $\alpha |\mathcal{D}_{\text{all}}| - |\mathcal{D}_{\text{train}}| = 0$, $\nabla_\theta \mathcal{L}_{\text{test}} = 0$. Otherwise, $\sin(\nabla_\theta \mathcal{L}_{\text{train}}, \nabla_\theta \mathcal{L}_{\text{test}}) = 0$. $\qquad\square$

**Lemma 10** (Equivalence when keeping training loss)**.**

$$\Delta = 0 \text{ and } \nabla_\theta \mathcal{L}_{test} = 0 \iff \nabla_\theta \mathcal{L}_{train} = \nabla_\theta \mathcal{L}_{all} = 0$$

*Proof.* To prove " $\implies$ ",

$$\Delta = 0 \implies \nabla_\theta \mathcal{L}_{\text{train}} \cdot \nabla_\theta \mathcal{L}_{\text{all}} = \nabla_\theta \mathcal{L}_{\text{train}} \cdot \left( \frac{|\mathcal{D}_{\text{train}}|}{|\mathcal{D}_{\text{all}}|} \nabla_\theta \mathcal{L}_{\text{train}} - \frac{|\mathcal{D}_{\text{test}}|}{|\mathcal{D}_{\text{train}}|} \nabla_\theta \mathcal{L}_{\text{test}} \right)$$

$$= \nabla_\theta \mathcal{L}_{\text{train}} \cdot \frac{|\mathcal{D}_{\text{all}}|}{|\mathcal{D}_{\text{train}}|} \nabla_\theta \mathcal{L}_{\text{train}} = 0 \implies \nabla_\theta \mathcal{L}_{\text{train}} = 0$$

$$\nabla_\theta \mathcal{L}_{\text{all}} = \frac{|\mathcal{D}_{\text{train}}|}{|\mathcal{D}_{\text{all}}|} \nabla_\theta \mathcal{L}_{\text{train}} + \frac{|\mathcal{D}_{\text{test}}|}{|\mathcal{D}_{\text{all}}|} \nabla_\theta \mathcal{L}_{\text{test}} = 0$$

To prove " $\impliedby$ ",

$$\nabla_\theta \mathcal{L}_{\text{train}} \cdot \nabla_\theta \mathcal{L}_{\text{all}} = 0 \implies \Delta = 0$$

$$\nabla_\theta \mathcal{L}_{\text{test}} = \frac{|\mathcal{D}_{\text{all}}|}{|\mathcal{D}_{\text{test}}|} \nabla_\theta \mathcal{L}_{\text{all}} - \frac{|\mathcal{D}_{\text{train}}|}{|\mathcal{D}_{\text{test}}|} \nabla_\theta \mathcal{L}_{\text{train}} = 0$$

$$\square$$

**Theorem 1** (Bias)**.** *There is a bias if and only if neither of the following holds.*

$$(A) \quad \nabla_\theta \mathcal{L}_{train} = \nabla_\theta \mathcal{L}_{all} = 0 \qquad\qquad (B) \quad \cos(\nabla_\theta \mathcal{L}_{train}, \nabla_\theta \mathcal{L}_{all}) = 1$$

*Proof.* By Lemma 1, we only need to prove a case is unbiased (Definition 4) if and only if condition (A) or (B) holds. By Proposition 3 and Lemma 10, (A) $\iff$ (5). By Proposition 1 and Lemma 9, (B) $\iff$ (4). Therefore, the conclusion holds. $\qquad\square$

### A.6 COROLLARY 1

**Corollary 1** (UDDR inequality).

$$UDDR(\mathcal{L}_{test}, \mathcal{L}_{train}, \nabla_\theta \mathcal{L}_{train}) \leq UDDR(\mathcal{L}_{test}, \mathcal{L}_{train}, \nabla_\theta \mathcal{L}_{all})$$

*The equal sign holds if and only if a case is unbiased.*

*Proof.* If $\Delta > 0$, UDDR equals DDR, so Theorem 1 applies.

If $\Delta < 0$, by Proposition 2,

$$a^T b < 0 < (a+b)^T b \implies \frac{a^T b}{|a^T a|} < 0 < \frac{(a+b)^T b}{|(a+b)^T a|}$$

Otherwise, the case $\Delta = 0$ in Lemma 2 applies. $\qquad\square$

### A.7 LEMMA 2

**Lemma 11** (UDDR difference). $\forall a, b \in \mathbb{R}^n$, *if* $(a+b)^T a < 0$, *then the following equation holds.*

$$\frac{(a+b)^T b}{|(a+b)^T a|} - \frac{a^T b}{|a^T a|} = \begin{cases} \frac{|b|^2 \sin^2(a,b) - 2(a+b)^T b}{(a+b)^T a}, & \text{If } b \neq 0 \\ 0, & \text{If } b = 0 \end{cases}$$

*Proof.*

$$(a+b)^T a \neq 0 \implies a \neq 0 \implies |a^T a| \neq 0$$

So $\frac{a^T b}{|a^T a|}$ is well-defined. If $b = 0$, the result is 0. If $b \neq 0$, we use Lemma 3.

$$\begin{aligned}
\frac{(a+b)^T b}{|(a+b)^T a|} - \frac{a^T b}{|a^T a|} &= -\frac{(a+b)^T b}{(a+b)^T a} - \frac{a^T b}{a^T a} \\
&= \frac{(a+b)^T b}{(a+b)^T a} - \frac{a^T b}{a^T a} - 2\frac{(a+b)^T b}{(a+b)^T a} \\
&= \frac{|b|^2 \sin^2(a,b)}{(a+b)^T a} - 2\frac{(a+b)^T b}{(a+b)^T a} \\
&= \frac{|b|^2 \sin^2(a,b) - 2(a+b)^T b}{(a+b)^T a}
\end{aligned}$$

$\qquad\square$

**Lemma 2** (UDDR derivation).

$$\Delta UDDR = UDDR(\mathcal{L}_{test}, \mathcal{L}_{train}, \nabla_\theta \mathcal{L}_{all}) - UDDR(\mathcal{L}_{test}, \mathcal{L}_{train}, \nabla_\theta \mathcal{L}_{train})$$

$$= \begin{cases} 0, & \text{if } \nabla_\theta \mathcal{L}_{test} = 0, \text{ otherwise} \\ \frac{|b|^2 \sin^2(a,b)}{(a+b)^T a}, & \text{if } \Delta > 0 \\ \frac{|b|^2 \sin^2(a,b) - 2(a+b)^T b}{(a+b)^T a}, & \text{if } \Delta < 0 \\ +\infty, & \text{if } \Delta = 0 \end{cases}$$

*Where* $a = |\mathcal{D}_{train}| \nabla_\theta \mathcal{L}_{train}$ *and* $b = |\mathcal{D}_{test}| \nabla_\theta \mathcal{L}_{test}$.

*Proof.*

$$|\mathcal{D}_{all}||\mathcal{D}_{train}|\Delta = (a+b)^T a \implies \Delta \text{ and } (a+b)^T a \text{ have the same sign.}$$
$$|\mathcal{D}_{test}|\nabla_\theta \mathcal{L}_{test} = b \implies \nabla_\theta \mathcal{L}_{test} \text{ and } b \text{ have the same sign.}$$

We use Definition 5.

If $\Delta > 0$, we use Lemma 3.

If $\Delta < 0$, we use Lemma 11.

If $\Delta = 0$, we use Lemma 8. If $\nabla_\theta \mathcal{L}_{\text{test}} \neq 0$, we have three possible cases.

$$a^T b < 0 < (a+b)^T b \implies \Delta\text{UDDR} = +\infty - -\infty = +\infty$$
$$a^T b < 0 \leq (a+b)^T b \implies \Delta\text{UDDR} = +\infty - 0 = +\infty$$
$$a^T b \leq 0 < (a+b)^T b \implies \Delta\text{UDDR} = 0 - -\infty = +\infty$$

Therefore, $\Delta\text{UDDR} = +\infty$.

If $\nabla_\theta \mathcal{L}_{\text{test}} = 0$, we have the following.

$$a^T b = 0 = (a+b)^T b \implies \Delta\text{UDDR} = 0 - 0 = 0$$

$\square$

## B   DETAILS IN EXPERIMENTS

### B.1   DATA

The CelebA dataset contains 202,599 face images of various celebrities[4]. The samples for each group are 89,931 "dark hair, female", 82,685 "dark hair, male", 28,234 "blond hair, female", 1,749 "blond hair, male". In the CFQ dataset[5], each MCD split has 95,743 training samples and 11,968 samples. For the NICO++ dataset, we aggregated foregrounds into five abstract classes, e.g., mammal and vehicle. It contains 72,176 image samples. For the Amazon review dataset, we randomly select 100,000 samples for each category, with a length limit of 100 tokens. For image data, each input element is linearly scaled to [-0.5, 0.5] for image input. Text inputs are preprocessed with the NLTK toolkit[6].

### B.2   SETTINGS

We use GeForce GTX 1080 or GeForce GTX 1050 Ti GPU for single GPU experiments. We use TensorFlow (Abadi et al., 2015) for implementation. The assets have public licenses. Each experiment requires around half an hour to two hours.

**Fully Connected Network**   The input shape is $28 \times 28 \times 3$ for the Fashion data and $64 \times 64 \times 3$ for the CelebA data. It is flattened to a vector. There are five fully connected layers. Each of them has 256 hidden nodes and ReLU activation. The output layer has Softmax activation. We use cross-entropy loss and Adam optimizer with a learning rate of 0.001. We also experiments with stochastic gradient descent (Appendix B.3). The batch size is 256, and we train 2,000 iterations. Evaluation at each step uses 1,000 samples.

**Convolutional Network**   The input shape is $64 \times 64 \times 3$. There are three convolutional layers. Each has $3 \times 3$ kernel size with 16 channels. Then the layer is flattened. We have two fully connected layers and ReLU activation. The first one has 32, and the second one has 16 nodes. The output layer has Softmax activation. We use cross-entropy loss and Adam optimizer with a learning rate of 0.001. We also experiments with stochastic gradient descent (Appendix B.3). The batch size is 256, and we train 2,000 iterations. Evaluation at each step uses 1,000 samples.

**Residual Network**   The input is the same as CNN. The model is the standard ResNet50 implementation. The output layer has Softmax activation. We use cross-entropy loss and Adam optimizer with a learning rate of 0.001. The batch size is 256, and we train 2,000 iterations. Evaluation at each step uses 256 samples. For CelebA data, the batch size is 128, and the test samples at each step are 128.

---

[4]Data is downloaded from `www.kaggle.com/datasets/jessicali9530/celeba-dataset`

[5]In CFQ, test samples may not all have zero probability in the training distribution. It is designed to maximize the difference between training and test datasets.

[6]`www.nltk.org`

**Vision Transformer** The input is the same as CNN. The model is the standard Vision Transformer implementation with five hidden layers. The hidden layer size is 64. The output layer has Softmax activation. We use cross-entropy loss and Adam optimizer with a learning rate of 0.001. The batch size is 256, and we train 2,000 iterations. Evaluation at each step uses 256 samples.

**LSTM** The input length is 100. The embedding size is 128. There are three stacked bidirectional LSTM layers, each with 64 hidden nodes for each direction. Then the output is flattened. The output layer is a fully-connected layer with Softmax activation. We use cross-entropy loss and Adam optimizer with a learning rate of 0.001. The batch size is 256, and we train 2,000 iterations. Evaluation at each step uses 256 samples. For the sequence prediction problem, hidden nodes are 128, the batch size is 32, and test samples at each step are 64.

**Transformer** The input is the same as that of LSTM. The embedding size is 256. There are three hidden groups. The hidden layer size is 256. The output is flattened. The output layer is a fully-connected layer with Softmax activation. We use cross-entropy loss and Adam optimizer with a learning rate of 0.001. The batch size is 256, and we train 2,000 iterations. Evaluation at each step uses 256 samples. For the sequence prediction problem, hidden nodes are 128, the batch size is 32, and test samples at each step are 64.

### B.3 MORE RESULTS

**CFQ MCD splits** We also have results for MCD2 and MCD3 splits (Figure 6).

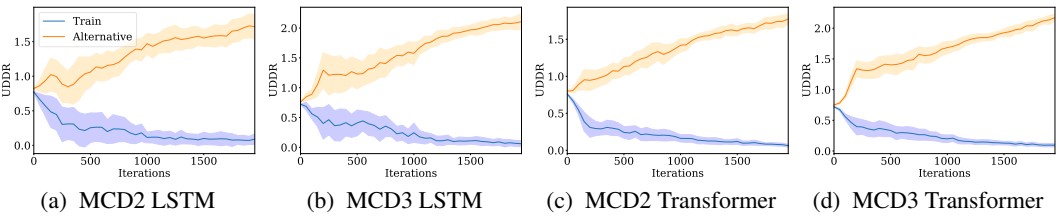

(a) MCD2 LSTM    (b) MCD3 LSTM    (c) MCD2 Transformer    (d) MCD3 Transformer

Figure 6: UDDR during training for more MCD splits. We plot in log scale while keeping signs.

**Stochastic gradient descent** We run additional experiments with vanilla stochastic gradient descent, which are consistent with the derivation in Section 2. We use the Fashion dataset for both fully connected and convolutional networks. The learning rate is 0.1, and other settings are the same as the corresponding multi-class classification tasks in the experiment section.

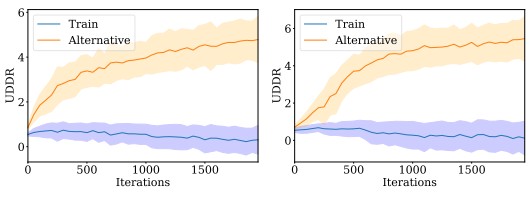

(a) Fully connected network (b) Convolutional network

Figure 7: UDDR during training for additional experiments. The score is significantly larger for alternative gradients than for training gradients in each experiment.

**Disentangled inputs** Unsupervised pre-training methods, such as variational auto encoder (VAE) methods, aim at learning disentangled representations. We assume that the disentangled representations $Z$ are perfectly learned, and we have identical hidden representation and output $Z = Y = Y_1, \ldots, Y_n$. Each $Y_i$ is a one-hot representation. We use the labels in the Fashion dataset, which has two 10-class outputs. We train fully connected models to predict output from the representations. The model and training settings are the same as the experiment section. The result in Figure 8 shows the bias, similar to previous experiments.

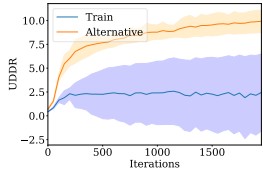

(a) Fully connected network

Figure 8: UDDR during training for disentangled experiments. The score is significantly larger for alternative gradients than for training gradients in each experiment.

## C  MORE DISCUSSION

### C.1  VARIANTS OF GRADIENTS

In practice, variants of gradient, e.g., Adam, are used to update parameters. Also, stochastic sampling may modify gradients. We discuss that the bias still exists under some conditions (Proposition 4). The proof is in the following subsection.

Suppose $\mathbf{u}_{\text{train}}$ is the modified training gradient $\nabla_\theta \mathcal{L}_{\text{train}}$ and $\mathbf{u}_{\text{test}}$ is the modified test gradient $\nabla_\theta \mathcal{L}_{\text{test}}$. We then define the modified alternative gradient as follows.

$$\mathbf{u}_{\text{all}} = \frac{1}{|\mathcal{D}_{\text{all}}|}(|\mathcal{D}_{\text{train}}|\mathbf{u}_{\text{train}} + |\mathcal{D}_{\text{test}}|\mathbf{u}_{\text{test}}) \in \mathbb{R}^n$$

Similar to $\Delta$, we have a threshold.

$$\Delta' = D(\mathcal{L}_{\text{train}}, \mathbf{u}_{\text{all}}) \in \mathbb{R}$$

We have the following conditions.

**Proposition 4** (Gradient variant). *Bias exists if one of the following conditions holds.*

(1)  $\Delta' > 0$   $D(\mathcal{L}_{train}, \mathbf{u}_{train}) > 0$   *and*   $\nabla_\theta \mathcal{L}_{test} \neq 0$   *and*   $\mathbf{u}_{test} \neq 0$   *and*
   $\cos(\nabla_\theta \mathcal{L}_{test}, \mathbf{u}_{train}) \cos(\nabla_\theta \mathcal{L}_{train}, \mathbf{u}_{test}) < \cos(\nabla_\theta \mathcal{L}_{train}, \mathbf{u}_{train}) \cos(\nabla_\theta \mathcal{L}_{test}, \mathbf{u}_{test})$

(2)  $\Delta' < 0$   $D(\mathcal{L}_{all}, \mathbf{u}_{train}) < 0$   *and*   $D(\mathcal{L}_{train}, \mathbf{u}_{train}) \geq 0$   *and*   $D(\mathcal{L}_{all}, \mathbf{u}_{all}) \geq 0$

(3)  $\Delta' = 0$   $D(\mathcal{L}_{all}, \mathbf{u}_{train}) \leq 0$   *and*   $D(\mathcal{L}_{train}, \mathbf{u}_{train}) \geq 0$   *and*   $D(\mathcal{L}_{all}, \mathbf{u}_{all}) \geq 0$   *and*
   *not*   $D(\mathcal{L}_{test}, \mathbf{u}_{train}) = D(\mathcal{L}_{test}, \mathbf{u}_{test}) = 0$

We mainly discuss (1) and (2) since (3) is a numerically rare case. A type of condition is the relation between a gradient and its modification, including $D(\mathcal{L}_{\text{train}}, \mathbf{u}_{\text{train}})$, $D(\mathcal{L}_{\text{test}}, \mathbf{u}_{\text{test}})$ and $D(\mathcal{L}_{\text{all}}, \mathbf{u}_{\text{all}})$. They are required to be equal to or above zero. Such conditions hold when the original and the modified gradients point in a close direction.

In (1), there are two types of cosine terms. The first type is the cosine value of an angle between a gradient and its modification, including $\cos(\nabla_\theta \mathcal{L}_{\text{train}}, \mathbf{u}_{\text{train}})$ and $\cos(\nabla_\theta \mathcal{L}_{\text{test}}, \mathbf{u}_{\text{test}})$. They are close to 1 if a gradient and its modification are close in direction. Another type is the cosine value of an angle between a training and a test gradient, one original and one modified, including $\cos(\nabla_\theta \mathcal{L}_{\text{test}}, \mathbf{u}_{\text{train}})$ and $\cos(\nabla_\theta \mathcal{L}_{\text{train}}, \mathbf{u}_{\text{test}})$. Their product is close to 0 if the training and test gradients are close to perpendicular. So the condition is likely to hold if the original and modified gradients are non-zero and close in direction, and the training and test gradients are close to perpendicular.

For (2), $D(\mathcal{L}_{\text{all}}, \mathbf{u}_{\text{train}})$ is likely to have the same sign as $\Delta'$ if the training and alternative gradients are close to their modifications, respectively (all non-zero). The other two conditions in (2) also hold in such cases.

### C.2  PROOFS

We use the following symbols for convenience in the proofs.

$$a = |\mathcal{D}_{\text{train}}|\nabla_\theta \mathcal{L}_{\text{train}} \qquad b = |\mathcal{D}_{\text{test}}|\nabla_\theta \mathcal{L}_{\text{test}} \qquad a' = |\mathcal{D}_{\text{train}}|\mathbf{u}_{\text{train}} \qquad b' = |\mathcal{D}_{\text{test}}|\mathbf{u}_{\text{test}}$$

**Lemma 12** (Difference). *$\forall a, a', b, b' \in \mathbb{R}^n$, if $(a' + b')^T a \neq 0$ and $a'^T a \neq 0$, then the following equation holds.*

$$\frac{(a' + b')^T b}{(a' + b')^T a} - \frac{a'^T b}{a'^T a} = \frac{|b||b'|(\cos(a, a')\cos(b, b') - \cos(a', b)\cos(a, b'))}{\cos(a, a')(a' + b')^T a}$$

*Proof.*

$$a'^T a \neq 0 \implies a' \neq 0, a \neq 0 \implies a'^T a = |a'||a|\cos(a, a') \neq 0 \implies \cos(a, a') \neq 0$$

$$\begin{aligned}
\frac{(a' + b')^T b}{(a' + b')^T a} - \frac{a'^T b}{a'^T a} &= \frac{(a' + b')^T b a'^T a - a'^T b (a' + b')^T a}{(a' + b')^T a a'^T a} \\
&= \frac{a'^T b a'^T a + b'^T b a'^T a - a'^T b a'^T a - a'^T b b'^T a}{(a' + b')^T a a'^T a} \\
&= \frac{b'^T b a'^T a - a'^T b b'^T a}{(a' + b')^T a a'^T a} \\
&= \frac{|b||b'|\cos(b, b')|a||a'|\cos(a, a') - |a'||b|\cos(a', b)|a||b'|\cos(a, b')}{|a||a'|\cos(a, a')(a' + b')^T a} \\
&= \frac{|b||b'|(\cos(a, a')\cos(b, b') - \cos(a', b)\cos(a, b'))}{\cos(a, a')(a' + b')^T a}
\end{aligned}$$

$\square$

**Lemma 13** (Reduce training loss). *Suppose $\Delta' > 0$.*

> *If*      $D(\mathcal{L}_{train}, \mathbf{u}_{train}) > 0$    *and*
>           $\cos(\nabla_\theta \mathcal{L}_{test}, \mathbf{u}_{train})\cos(\nabla_\theta \mathcal{L}_{train}, \mathbf{u}_{test}) \leq \cos(\nabla_\theta \mathcal{L}_{train}, \mathbf{u}_{train})\cos(\nabla_\theta \mathcal{L}_{test}, \mathbf{u}_{test})$
> *Then*    $DDR(\mathcal{L}_{text}, \mathcal{L}_{train}, \mathbf{u}_{train}) \leq DDR(\mathcal{L}_{text}, \mathcal{L}_{train}, \mathbf{u}_{all})$

*The equal sign holds if and only if*

> $\nabla_\theta \mathcal{L}_{test} = 0$ *or* $\mathbf{u}_{test} = 0$ *or*
> $\cos(\nabla_\theta \mathcal{L}_{train}, \mathbf{u}_{train})\cos(\nabla_\theta \mathcal{L}_{test}, \mathbf{u}_{test}) = \cos(\nabla_\theta \mathcal{L}_{test}, \mathbf{u}_{train})\cos(\nabla_\theta \mathcal{L}_{train}, \mathbf{u}_{test})$

*Proof.* Suppose $\Delta' > 0$. We use Lemma 12.

If $b = 0$ or $b' = 0$, $DDR(\mathcal{L}_{text}, \mathcal{L}_{train}, \mathbf{u}_{all}) - DDR(\mathcal{L}_{text}, \mathcal{L}_{train}, \mathbf{u}_{train}) = 0$.

Otherwise,

$$D(\mathcal{L}_{train}, \mathbf{u}_{train}) > 0 \implies \cos(a, a') > 0$$

So we have

$$\begin{aligned}
&DDR(\mathcal{L}_{text}, \mathcal{L}_{train}, \mathbf{u}_{all}) - DDR(\mathcal{L}_{text}, \mathcal{L}_{train}, \mathbf{u}_{train}) \\
=&\alpha(\cos(\nabla_\theta \mathcal{L}_{train}, \mathbf{u}_{train})\cos(\nabla_\theta \mathcal{L}_{test}, \mathbf{u}_{test}) - \cos(\nabla_\theta \mathcal{L}_{test}, \mathbf{u}_{train})\cos(\nabla_\theta \mathcal{L}_{train}, \mathbf{u}_{test})) \geq 0
\end{aligned}$$

Where $\alpha > 0$. So the result follows. The equal sign holds if and only if

> $\nabla_\theta \mathcal{L}_{test} = 0$ or $\mathbf{u}_{test} = 0$ or
> $\cos(\nabla_\theta \mathcal{L}_{train}, \mathbf{u}_{train})\cos(\nabla_\theta \mathcal{L}_{test}, \mathbf{u}_{test}) = \cos(\nabla_\theta \mathcal{L}_{test}, \mathbf{u}_{train})\cos(\nabla_\theta \mathcal{L}_{train}, \mathbf{u}_{test})$

$\square$

**Lemma 14** (Increase training loss). *Suppose $\Delta' < 0$.*

> *If*      $D(\mathcal{L}_{all}, \mathbf{u}_{train}) < 0$    *and*    $D(\mathcal{L}_{train}, \mathbf{u}_{train}) \geq 0$    *and*    $D(\mathcal{L}_{all}, \mathbf{u}_{all}) \geq 0$
> *Then*    $D(\mathcal{L}_{test}, \mathbf{u}_{train}) < 0 < D(\mathcal{L}_{test}, \mathbf{u}_{all})$

*Proof.* To prove $0 < D(\mathcal{L}_{\text{test}}, \mathbf{u}_{\text{all}})$,

$$\Delta' < 0 \implies (a' + b')^T a < 0$$
$$D(\mathcal{L}_{\text{all}}, \mathbf{u}_{\text{all}}) \geq 0 \implies (a' + b')^T (a + b) \geq 0$$

We then have

$$(a' + b')^T b > (a' + b')^T a + (a' + b')^T b = (a' + b')^T (a + b) \geq 0$$

To prove $D(\mathcal{L}_{\text{test}}, \mathbf{u}_{\text{train}}) < 0$,

$$D(\mathcal{L}_{\text{all}}, \mathbf{u}_{\text{train}}) < 0 \implies (a + b)^T a' < 0$$
$$D(\mathcal{L}_{\text{train}}, \mathbf{u}_{\text{train}}) \geq 0 \implies a^T a' \geq 0$$

We then have

$$a'^T b \leq a'^T b + a'^T a = (a + b)^T a' < 0$$

Therefore,

$$D(\mathcal{L}_{\text{test}}, \mathbf{u}_{\text{train}}) < 0 < D(\mathcal{L}_{\text{test}}, \mathbf{u}_{\text{all}})$$

$\square$

**Lemma 15** (Keep training loss). *Suppose $\Delta' = 0$.*

$$\text{If} \qquad D(\mathcal{L}_{all}, \mathbf{u}_{train}) \leq 0 \quad \text{and} \quad D(\mathcal{L}_{train}, \mathbf{u}_{train}) \geq 0 \quad \text{and} \quad D(\mathcal{L}_{all}, \mathbf{u}_{all}) \geq 0$$
$$\text{Then} \qquad D(\mathcal{L}_{test}, \mathbf{u}_{train}) \leq 0 \leq D(\mathcal{L}_{test}, \mathbf{u}_{all})$$

*The two equal signs hold simultaneously if and only if $D(\mathcal{L}_{test}, \mathbf{u}_{train}) = D(\mathcal{L}_{test}, \mathbf{u}_{test}) = 0$.*

*Proof.* To prove $0 \leq D(\mathcal{L}_{\text{test}}, \mathbf{u}_{\text{all}})$,

$$\Delta' = 0 \implies (a' + b')^T a = 0$$
$$D(\mathcal{L}_{\text{all}}, \mathbf{u}_{\text{all}}) \geq 0 \implies (a' + b')^T (a + b) \geq 0$$

We then have

$$(a' + b')^T b = (a' + b')^T a + (a' + b')^T b = (a' + b')^T (a + b) \geq 0$$

To prove $D(\mathcal{L}_{\text{test}}, \mathbf{u}_{\text{train}}) \leq 0$,

$$D(\mathcal{L}_{\text{all}}, \mathbf{u}_{\text{train}}) \leq 0 \implies (a + b)^T a' \leq 0$$
$$D(\mathcal{L}_{\text{train}}, \mathbf{u}_{\text{train}}) \geq 0 \implies a^T a' \geq 0$$

We then have

$$a'^T b \leq a'^T b + a'^T a = (a + b)^T a' \leq 0$$

Therefore,

$$D(\mathcal{L}_{\text{test}}, \mathbf{u}_{\text{train}}) \leq 0 \leq D(\mathcal{L}_{\text{test}}, \mathbf{u}_{\text{all}})$$

To prove the equal signs.

$$b^T a' = b^T b' = 0 \implies b^T (a' + b') = b^T a' + b^T b' = 0$$
$$b^T a' = 0 = b^T (a' + b') \implies b^T b' = b^T (a' + b') - b^T a' = 0$$

Therefore,

$$D(\mathcal{L}_{\text{test}}, \mathbf{u}_{\text{train}}) = 0 = D(\mathcal{L}_{\text{test}}, \mathbf{u}_{\text{all}}) \iff D(\mathcal{L}_{\text{test}}, \mathbf{u}_{\text{train}}) = D(\mathcal{L}_{\text{test}}, \mathbf{u}_{\text{test}}) = 0$$

$\square$

**Proposition 4** (Gradient variant). *Bias exists if one of the following conditions holds.*

(1) $\Delta' > 0$   $D(\mathcal{L}_{train}, \mathbf{u}_{train}) > 0$   *and*   $\nabla_\theta \mathcal{L}_{test} \neq 0$   *and*   $\mathbf{u}_{test} \neq 0$   *and*
$$\cos(\nabla_\theta \mathcal{L}_{test}, \mathbf{u}_{train}) \cos(\nabla_\theta \mathcal{L}_{train}, \mathbf{u}_{test}) < \cos(\nabla_\theta \mathcal{L}_{train}, \mathbf{u}_{train}) \cos(\nabla_\theta \mathcal{L}_{test}, \mathbf{u}_{test})$$

(2) $\Delta' < 0$   $D(\mathcal{L}_{all}, \mathbf{u}_{train}) < 0$   *and*   $D(\mathcal{L}_{train}, \mathbf{u}_{train}) \geq 0$   *and*   $D(\mathcal{L}_{all}, \mathbf{u}_{all}) \geq 0$

(3) $\Delta' = 0$   $D(\mathcal{L}_{all}, \mathbf{u}_{train}) \leq 0$   *and*   $D(\mathcal{L}_{train}, \mathbf{u}_{train}) \geq 0$   *and*   $D(\mathcal{L}_{all}, \mathbf{u}_{all}) \geq 0$   *and*
$$not \quad D(\mathcal{L}_{test}, \mathbf{u}_{train}) = D(\mathcal{L}_{test}, \mathbf{u}_{test}) = 0$$

*Proof.* (1), (2), and (3) hold because of Lemma 13, Lemma 14, and Lemma 15, respectively.    $\square$

