# OpenReview forum: "On the Relation between Gradient Directions and Systematic Generalization"
_ICLR.cc/2024/Conference — ICLR 2024 Conference Withdrawn Submission_

### Official Review · Reviewer_jCQf · 2023-10-28

**Soundness:** 2 fair
**Presentation:** 1 poor
**Contribution:** 2 fair
**Rating:** 3
**Confidence:** 4

**Summary:**

The paper studies systematic generalization problems from the perspective of gradient directions. Based on the analysis of gradients on different distributions, the paper points out a bias that the training gradient is less efficient than another alternative gradient (the gradient on the ground-truth data distribution for all samples). Such a bias is experimentally verified on many classical deep neural networks, like CNN, LSTM, ViT, etc.

Although the idea of this paper is novel to me, the analysis and presentation of the paper are not good. It is hard for me to follow the paper, and I cannot see how the community can apply the proposed analysis to improve systematic generalization.

**Strengths:**

See the summary part.

**Weaknesses:**

Although studying the learning dynamics from the perspective of the gradient on different distributions (i.e., training, test, and overall) is novel, the paper lacks persuasive results and good presentations. It is hard for me to conclude the contribution of this paper. I have the following two main concerns.

First, the systematic generalization (sys-gen) is not defined in this paper. What’s the difference between sys-gen and OOD problem? Are there any constraints on the differences between training and test distributions? What are the differences and similarities between training and test distributions in this setting?

Second, it is hard to see any potential of the provided analysis. The paper proposes several concepts based on gradient directions and then proposes a metric named UDDR. How is the UDDR gap related to systematic generalization, and how could we improve the systematic generalization performance based on the findings?

**Questions:**

1. Alternative direction is not a good term in my opinion. I guess this direction means the direction of the ground-truth data distribution.

2. In definition 3, “alternative gradient reduces the training loss”. I think this type of claim is a little weird. Because in practice, learning can only reduce the training loss, while the gradient for all data is unobservable. Better to say “training gradient reduces the alternative loss”. Or just call this "alternative gradient" an "oracle gradient", which means the optimal but inaccessible correct gradient.

3. Many of the concepts and definitions in this paper are based on vector inner produce, e.g., definition 3, 4, proposition 1, 2, etc. It is very hard to remember what they are discussing without a clear visualization. I think it would be helpful to visualize these concepts using a series of figures (similar to Figure 2) somewhere in the paper.

---

> ### Author Response · Authors · 2023-11-18
> **Rebuttal**
>
> > Q1. Although studying the learning dynamics from the perspective of the gradient on different distributions
> (i.e., training, test, and overall) is novel, the paper lacks persuasive results and good presentations. It is
> hard for me to conclude the contribution of this paper. I have the following two main concerns.
>
> A1. The main contribution is the proposed formulation of treating reducible training loss as a resource and
> compute efficiency.
>
> > Q2. First, the systematic generalization (sys-gen) is not defined in this paper. What’s the difference
> between sys-gen and OOD problem? Are there any constraints on the differences between training and
> test distributions? What are the differences and similarities between training and test distributions in this
> setting?
>
> A2. In an OOD problem, some test samples have zero probability in training distribution. The sys-gen is a
> type of OOD problem, and it has a new combination of seen factors in the test. Datasets in the experiment
> sections are examples. Since it has the new combination, the test samples are likely to be distant from the
> support of training distribution (which may not be true for all OOD problems). So, the training and the
> alternative gradients are not likely to have the same direction. This assumption on the difference in direction
> is important to apply the theorem. So, in this paper, the main characteristic of sys-gen is that the training
> and the test distributions have large differences.
>
> > Q3. Second, it is hard to see any potential of the provided analysis. The paper proposes several concepts
> based on gradient directions and then proposes a metric named UDDR. How is the UDDR gap related to
> systematic generalization, and how could we improve the systematic generalization performance based on
> the findings?
>
> A3. UDDR is the efficiency of consuming training loss as a resource. When the efficiency is low, the test
> losses are less reduced, so there is a bias not to achieve systematic generalization. This paper discusses
> standard architectures and explains the difficulties for them. It provides a theoretical explanation of the
> bias, which is often empirically observed. It is a theoretical basis for why we should avoid using the standard
> networks. When we design new architecture or regularization for sys-gen, it says we should consider whether
> it avoids the bias.
>
> A4. Also, thank you for the writing and figure suggestions. We will improve them.

---

> > ### Comment · Reviewer_jCQf · 2023-11-19
> >
> > Thanks very much for the response, I will keep my original score.

---

### Official Review · Reviewer_ymSN · 2023-10-30

**Soundness:** 2 fair
**Presentation:** 1 poor
**Contribution:** 1 poor
**Rating:** 3
**Confidence:** 3

**Summary:**

The paper proposes a measure of out-of-distribution generalization bias based on a \textit{total} and \textit{training} gradient (cosine) similarity. After introducing definitions and lemmas, authors show (Theorem 1) that, letting alone degenerate case of zero gradients, the systematic generalization bias exists whenever  \textit{total} and \textit{training} gradients point in DIFFERENT direction, i.e., their cosine similarity is < 1. The claim is (partially) supported by experiments on the most popular deep learning architectures incl. transformers, LSTM and ResNet.

**Strengths:**

+ Well written Introduction section
+ Addressing a timely and needed research topic of systematic generalisation of deep learning models
+ Thorough build up of definitions and lemmas leading to the main Theorem 1 and related Propositions and Corollaries

**Weaknesses:**

While claim of Theorem 1 seems almost trivial from Definitions 1-4 (bias), the most of the paper covers formalism of this relation. But it rather misses explaining why would such a "local" step-wise definition of systematic generalization bias (see bellow for what is meant by "local") capture a "global",i.e., after all gradient descent steps are done, o.o.d. generalization gap. In my opinion the paper in its current form has made interesting initial steps but has not succeeded in showing the proposed measure of bias is a "good" one (in a sense described bellow).

- While experimental setup is described at length, the summary of experiments is only one paragraph long and lacking needed details. More over, Contributions, bullet 3 on page 2, it is claimed: “Experiments validate the result and demonstrate a bias in standard deep learning models ”.

What is meant by “validate” exactly? The fact that there is generalization gap present in the most of DL models is generally known. If authors want to show that UDDR gap is good measure of it then they should "validate" not only UDDR gap =>"bad systematic generalization", but as well a contraposition the statement, i.e.,  "good o.o.d. generalization => low UDDR gap". The experiments do not show any such example to my knowledge.

- “We also discuss that systematic generalization requires a network decomposed to sub-networks, each with a seen test inputs. ” This is very interesting suggestion and possible research direction that would deserve a bit more comments perhaps. The section 4.2 only provides very brief and, to me, incomprehensible comments, however. Especially the Requirement on page 8, “ Systematic generalization requires that a model can be decomposed into sub- networks, each with seen test inputs. ”. For instance what is a “subnetwork with seen test inputs?”

- Proposed method of DDR (and D()) measures step-wise bias, i.e., a "local" bias at a given step of a gradient descent. How does this local bias relate to a final, "global", systematic generalization gap? Can there be two different pathways leading to the same model/result? Why not?

**Questions:**

In addition to questions raised in "Weaknesses" section.

- Contributions: Claim No. 3 on page 2: “Experiments validate the result and demonstrate a bias in standard deep learning models ”. What is meant by “validate” exactly? The fact that there is generalization gap present in the most of DL models is generally known. If authors want to show that UDDR gap is good measure of it then they should "validate" not only UDDR gap =>"bad systematic generalization", but as well a contraposition the statement, i.e.,  "good o.o.d. generalization => low UDDR gap". The experiments do not show any such example to my knowledge.

- What is “sin” in Prop 1? Please add am explanatory note …

- Fig 4. UDDR of exactly what is depicted? It has three arguments … is if UDDR(test, train, all)?

-(Q): Fig 4, 5 Did both networks converge to same or different optima? Could you add these results?

More generally, how does proposed approach deal with stochastic gradient descent with noisy gradient and possibly several paths leading to the same solution? The appendix treats this very briefly and does not answer the question in my opinion.

---

> ### Author Response · Authors · 2023-11-18
> **Rebuttal 1/2**
>
> > Q1. While claim of Theorem 1 seems almost trivial from Definitions 1-4 (bias), the most of the paper
> covers formalism of this relation.
>
> A1. The point is that definitions 3 and 4 may not cover all possible cases. So, it looks trivial because
> Definition 4 contains only rare cases, so the biased cases (Definition 3) seem to dominate spontaneously.
> However, we need Proposition 1,2,3 to prove they are the only cases (Lemma 1), and the derivation is trivial.
>
> > Q2. Proposed method of DDR (and D()) measures step-wise bias, i.e., a ”local” bias at a given step of
> a gradient descent. How does this local bias relate to a final, ”global”, systematic generalization gap? Can
> there be two different pathways leading to the same model/result? Why not? How does proposed approach
> deal with possibly several paths leading to the same solution?
>
> A2. The questions are about the relation between local and global behaviors. Multiple paths can lead to the
> same solution. However, a path following training gradients is less likely to achieve a solution of systematic
> generalization. The point is that the theorem compares two efficiencies at the same point in the parameter
> space. So, it fits to study whether a pathway can reduce enough test loss, but it does not fit to directly
> compare different pathways.
>
> Two different pathways can lead to the same model. Suppose different pathways P and Q. Q follows the
> alternative gradient at each step and is an oracle path. When the pathways deviate, P has less efficiency. If
> the two pathways lead to the same result, they have the same final test loss. It means P will catch up with Q
> after deviating. This is possible because the efficiency is compared between two gradients at the same point
> in the parameter space. Since P and Q are already apart, they are not at the same point, so the theorem
> does not apply.
>
> We then look at the case that P follows the training pathway. This paper assumes that the training and the
> alternative gradients do not point in the same direction for standard architectures in systematic generalization
> problems. The assumption and the theorem apply to each step in P. It means each step in P has a bias in
> efficiency, compared with an alternative gradient that leads to the final test loss of the Q pathway. So, the
> final test loss of the P pathway unlikely achieves that of the Q pathway. It also means that the P and Q
> pathways have different final models. Also, the experiments show that the efficiency gap will likely increase
> during training, suggesting the P is not likely to catch up.
>
> We aim to develop a theory for local behavior. The gradient is defined locally, so developing a theory in
> local space is more natural. The training process may be unstable since a small difference may cause a large
> difference after multiple steps, which makes it hard to develop a strict theory. In analogous, gradients are
> used to train models, though it does not guarantee global optimum.
>
> > Q3. While experimental setup is described at length, the summary of experiments is only one paragraph
> long and lacking needed details.
>
> A3. Thank you, and we would like to improve the result summary. The purposes and settings of experiments
> are stated before it, so the summary mainly needs to check whether the results meet the expectations. Other
> details of the experiments are in Appendix B. We assume that the training and the alternative gradients have
> different gradient directions in each training step. The result shows the assumption holds in the experiments.
>
> > Q4. More over, Contributions, bullet 3 on page 2, it is claimed: “Experiments validate the result and
> demonstrate a bias in standard deep learning models ”. What is meant by “validate” exactly?
>
> A4. “Validate” means providing common examples to check that the derived result is correct. It helps to
> improve the confidence of the proofs, especially when it is long.
>
> > Q5. The fact that there is generalization gap present in the most of DL models is generally known. If
> authors want to show that UDDR gap is good measure of it then they should ”validate” not only UDDR
> gap => ”bad systematic generalization”, but as well a contraposition the statement, i.e., ”good o.o.d.
> generalization => low UDDR gap”. The experiments do not show any such example to my knowledge.
>
> A5. The purpose of this paper is to provide a theoretical explanation of empirical observation that standard
> deep learning models often do not generalize systematically. We mainly focus on local behaviors. It is a
> good suggestion to test the contraposition statement. It is an additional support. In this study, we do not
> find a standard architecture trained to achieve the generalization ability, so we do not have examples for the
> test. We like to think how we can do it.

---

> ### Author Response · Authors · 2023-11-18
> **Rebuttal 2/2**
>
> > Q6. “We also discuss that systematic generalization requires a network decomposed to sub-networks,
> each with a seen test inputs.” This is very interesting suggestion and possible research direction that would
> deserve a bit more comments perhaps. The section 4.2 only provides very brief and, to me, incomprehensible
> comments, however. Especially the Requirement on page 8, “Systematic generalization requires that a model
> can be decomposed into sub-networks, each with seen test inputs. ”. For instance what is a “subnetwork
> with seen test inputs?”
>
> A6. An example of seen test input is that a subnetwork is a patch in a convolutional layer. Though the whole
> input for a test sample is unseen, a patch may be seen. It may also be a part of a hidden representation.
> We would like to add more comments in the discussion section.
>
> > Q7. What is “sin” in Prop 1? Please add am explanatory note.
>
> Q7. sin is sine trigonometric function.
>
> > Q8. Fig 4. UDDR of exactly what is depicted? It has three arguments . . . is if UDDR(test, train, all)?
>
> A8. UDDR is a scalar value corresponding to the efficiency of consuming training loss. Figure 4 depicts
> UDDR values in each iteration (step) of the training.
>
> > Q9. Fig 4, 5 Did both networks converge to same or different optima? Could you add these results?
>
> A9. The following tables show the training and the test losses in the trained models in each experiment.
> In each experiment, the losses have a significant difference, indicating the models do not achieve the generalization.
> It also means the trained parameters differ from those that enable the generalization.
> Note that rows may have different datasets (e.g., image or text), so they are not comparable.
>
> Train and test losses on trained models for experiments in Figure 4 and Figure 5.
> | Figure 4 | Train loss | Test loss |
> | ----------- | ----------- | ----------- |
> | Fully connected network | 0.11 $\pm$ 0.01 | 2.33 $\pm$ 0.08 |
> | Convolutional network   | 0.03 $\pm$ 0.01 | 6.22 $\pm$ 0.23 |
> | Residual Network        | 0.25 $\pm$ 0.09 | 3.53 $\pm$ 0.46 |
> | Vision Transformer      | 0.28 $\pm$ 0.02 | 2.28 $\pm$ 0.07 |
> | LSTM                    | 0.26 $\pm$ 0.01 | 2.49 $\pm$ 0.02 |
> | Transformer             | 0.19 $\pm$ 0.01 | 3.01 $\pm$ 0.10 |
>
> | Figure 5 | Train loss | Test loss |
> | ----------- | ----------- | ----------- |
> | Fully connected network | 0.01 $\pm$ 0.00 | 7.19 $\pm$ 0.66 |
> | Convolutional network   | 0.00 $\pm$ 0.00 | 7.95 $\pm$ 1.54 |
> | Residual Network        | 0.03 $\pm$ 0.02 | 3.78 $\pm$ 0.55 |
> | Vision Transformer      | 0.00 $\pm$ 0.01 | 5.34 $\pm$ 0.55 |
> | LSTM                    | 0.36 $\pm$ 0.01 | 0.78 $\pm$ 0.04 |
> | Transformer             | 0.41 $\pm$ 0.01 | 0.85 $\pm$ 0.03 |
>
> > Q10. More generally, how does proposed approach deal with stochastic gradient descent with noisy
> gradient? The appendix treats this very briefly and does not answer the question in my opinion.
>
> A10. Appendix C.1 discusses variants of gradients, and stochastic gradient can be regarded as one of them.

---

> ### Comment · Reviewer_ymSN · 2023-11-19
>
> I thank authors for their responses. My concerns have not been alleviated, I am afraid. The most pressing are Q2 and Q5.
> As for Q2.
>
> > from A2: "... a path following training gradients is less likely to achieve a solution of systematic generalization ...". Where is it shown in the paper?
>
> I try to clarify Q2 a bit more. Consider convex non-symetrical problem (eigenvalues of Hessian are not equal). Then natural (preconditioned by inverse Hessian) vs. vanilla gradient descent, starting (by choice) and ending (by construction) at the same solution, will take different paths. Whatever oracle path is (could even be one of them) all solutions coincide and have same properties. Relatively natural and vanilla GD are inferior to each other as their efficiency biases to oracle path differ (due to preconditioning). But at the same time they are not less likely to achieve a solution of systematic generalization (claimed above).
>
> Could authors clarify this through prism of the paper?
>
> Ad A5.)  Especially because the paper "... provides a theoretical explanation..." and since "A =>B" is equivalent to "B' => A'", where prime " ' " denotes logical negation, the contraposition mentioned in Q5 has to hold. Could authors show it in the "convex" settings, as the one above, perhaps?

---

> > ### Author Response · Authors · 2023-11-22
> >
> > Thank you for the detailed update.
> > We would like to have the following answers.
> >
> > > from A2: ”... a path following training gradients is less likely to achieve a solution of systematic generalization ...”. Where is it shown in the paper?
> >
> > It is not shown in the paper. The paper focuses on local properties, so it is an extended discussion.
> >
> > > I try to clarify Q2 a bit more. Consider convex non-symetrical problem (eigenvalues of Hessian are not
> > equal). Then natural (preconditioned by inverse Hessian) vs. vanilla gradient descent, starting (by choice)
> > and ending (by construction) at the same solution, will take different paths. Whatever oracle path is (could
> > even be one of them) all solutions coincide and have same properties. Relatively natural and vanilla GD
> > are inferior to each other as their efficiency biases to oracle path differ (due to preconditioning). But at the
> > same time they are not less likely to achieve a solution of systematic generalization (claimed above). Could
> > authors clarify this through prism of the paper?
> >
> > The point is that the loss functions for training data and all data are different, so their solutions can be
> > different (however, this example has the same solution for the two gradients). So, we would like to discuss the
> > training and the alternative gradient pathways and analyze a case similar to the convex example where the
> > two gradient pathways end at the same point. Developing a theory for the entire training process may require
> > addressing additional problems. For example, we may need an assumption to convert a local advantage (of
> > the alternative gradient as stated in the theorem) to a global one.
> >
> > The statement for the global property is the following: If the training and the alternative gradients deviate
> > at a step on the training gradient pathway, the training gradient pathway does not reduce enough test loss
> > (for the all-data solution) at the end.
> >
> > (Assumption) Suppose we have two nearby points, A and B, in parameter space. They have the same
> > training loss, and A has a lower test loss than B. Starting from A, we follow the training gradient pathway
> > to achieve A’. Starting from B, we also follow the training gradient pathway to achieve B’. If A’ and B’ have
> > the same training loss, then A’ has a lower test loss than B’.
> >
> > The assumption says that training pathways maintain the comparison of test loss given the same training
> > loss. It means that if we could take an alternative gradient at a step in the training gradient pathway,
> > the final test loss is lower when training finishes. So, by repeatedly applying the assumption, the final test
> > loss is even lower if we follow the alternative gradient pathway starting from the step (and ending at an
> > all-data solution). So, the statement holds. This assumption does not hold in the convex example because
> > all pathways have the same final test loss.
> >
> > > Ad A5.) Especially because the paper ”... provides a theoretical explanation...” and since ”A => B” is
> > equivalent to ”B’ => A’”, where prime ” ’ ” denotes logical negation, the contraposition mentioned in Q5
> > has to hold. Could authors show it in the ”convex” settings, as the one above, perhaps?
> >
> > The contraposition is the following: If the training gradient pathway reduces enough test loss (for the all-data
> > solution) at the end, the training and the alternative gradients do not deviate at any step on the training
> > gradient pathway. In the convex example, since a training pathway reduces enough test loss (because it ends
> > at the all-data solution), the training and the alternative gradients do not deviate at any step.

---

> > > ### Comment · Reviewer_ymSN · 2023-11-23
> > >
> > > Thanks authors for their responses. Unfortunately, they have not alleviated raised concerns.
> > >
> > > For instance, the response to A5 regarding contraposition in the rebuttal above"...the training and the alternative gradients do not deviate at any step." Considering GD and natural GD (gradient multiplied by inverse of Hessian) in the convex case, these two gradient directions do differ at every step (for non-symmetric Hessian) but still converge to same train and test error ...
> > >
> > > I keep my rating. Thank you.

---

> > > > ### Author Response · Authors · 2023-11-23
> > > >
> > > > Thanks. Here is a quick answer. In the case of the A5 with GD and natural GD, the (Assumption) does not hold.

---

### Official Review · Reviewer_SN1k · 2023-11-01

**Soundness:** 2 fair
**Presentation:** 2 fair
**Contribution:** 2 fair
**Rating:** 3
**Confidence:** 4

**Summary:**

This work explores how the alignment of the parameter update of a neural network followed by gradient descent with that of the gradient which aligns with the true (all data distribution) gradient. A number of cases are considered, which provable cover all possible outcomes, and it is shown that standard deep learning models are predominantly biased away from systematic generalisation. This argument is supported empirically for a range of practical models and tasks.

**Strengths:**

## Originality
The notion of treating the gradient update step as a resource which must be allocated is indeed interesting and an intuitive interpretation of events.

## Quality
The experimental design of Section 3 does support the main claim of the theory that neural networks are inherently biased away from generalisation. In addition the experiments cover a wide range of models that are of practical interest to the community. This makes the results interesting and useful. The claims which are made from these experimental results also appear accurate. On the theory side, definitions are clearly stated and used consistently throughout and propositions are formal but also explained clearly.

## Clarity
The work clearly describes the hypothesis. Figures are clear, interpretable and relate easily to the text.

## Significance
This work aims to address an interesting problem, and importantly does so for a broad range of practically useful models. Thus, the proposed finding of explaining why this broad range of models fail to systematically generalise could be of high impact. I do have some concerns on whether the work fully realises its intended purpose which I discuss in the Weaknesses section below. This limits the potentially high impact and significance of this work.

**Weaknesses:**

## Quality
My main issue with this work is that it does not seem to address systematic generalisation, but generalisation more broadly. A primary purpose of systematicity and compositional generalisation is to break the task into smaller pieces which are then learned (the modules specialise to the subtask) and composed later. By definition learning these smaller pieces is not the same as learning the entire data distribution. This seems at odds with this work's proposed theory that how far a network deviates from the all gradient direction is an indicator of systematicity. For example, learning a module which identifies the colour red, another which identifies cars and then learning to use both modules to identify red cars - as would be the case with neural module networks [1] - would follow extremely different gradient directions than if a similar network trained to directly identify all red cars. Put another way, to learn a disentangled representation [2] (a stronger condition than systematicity) you would need to follow a different gradient than if you learned the ground-truth mapping directly. Even from a linguistics perspective, seeing only a subset of data and this being used to learn something different from memorizing all of language is a foundational idea in systematicity [3]. Finally, more recent theory even explicitly makes the distinction between generalising because the network has seen enough data and generalising because the network decomposed the problem and learned a solution with an entirely different rank [4]. This last point is from a paper which is only a year old - and so open to debate - but this work would need to at least show how its definition of systematicity aligns with these prior notions. So for example the line "The alternative gradient is computed from all data, leading to systematic generalization" is just at odds with our notions of systematicity. This work, is of interest to generalisation broadly however, just not systematic generalisation as far as I can tell. Also the mention of "seen test inputs" is confusing and I don't know what this is meant to be referring to. But by definition test inputs are unseen.

## Clarity
On the point of clarity, there are some statements in this work which do not make sense or appear out of context. Examples are "Also, deep learning does not require many task-specific designs for specific tasks", "Some standard networks, such as ... work well in i.i.d. settings", "To keep the advantage, we  discuss whether standard deep learning models achieve systematic generalization", "The condition $\Delta=0$ is rare to hold because it requires an equation to hold" and "Both (A) and (B) contain equal signs, which are generally difficult to hold". Hopefully these examples will guide a general clean up of the writing.

A few more quicker points and concerns on clarity are the following. The last paragraph of page 2 where the notation is introduced is also not clear and introduces more notation than necessary. Why are $u$ and $h$ defined here and why have $u$ if $x$ already denotes input vectors? Similarly, $D(f,u)$ is defined and includes a case for if $u=0$ where on the top of the same page it is stated that $u \neq 0$. $\Delta$ is overall unhelpful as it obscures comparison with the other uses of  the $D(\cdot,\cdot)$ function. Definition 3 and 4 could be merged with Propositions 1, 2 and 3 since the propositions follow immediately from the definitions. Proposition 2 and 3 could also just be combined since the two cases are practically identical. DDR is also only used for two of five cases and so appears to be another function needlessly defined which just obscures the comparison of various cases. Also, why not use UDDR from the beginning? The captions for Figure 3 and Table 1 should be improved and clearly state why we should care about these datasets, how they are used and why they relate to systematicity.

[1] Andreas, Jacob, et al. "Neural module networks." Proceedings of the IEEE conference on computer vision and pattern recognition. 2016. \
[2] Locatello, Francesco, et al. "Challenging common assumptions in the unsupervised learning of disentangled representations." international conference on machine learning. PMLR, 2019. \
[3] Hadley, Robert F. "Systematicity in connectionist language learning." Mind & Language 9.3 (1994): 247-272. \
[4] Jarvis, Devon, et al. "On The Specialization of Neural Modules." The Eleventh International Conference on Learning Representations. 2022.

**Questions:**

I have asked a number of questions and raised some concerns in the Weaknesses section where they naturally came up. I do not currently have any further questions for this section but would appreciate if these early questions were addressed.

---

> ### Author Response · Authors · 2023-11-18
> **Rebuttal 1/2**
>
> > ### Quality
> > Q1. My main issue with this work is that it does not seem to address systematic generalisation, but
> generalisation more broadly. A primary purpose of systematicity and compositional generalisation is to break
> the task into smaller pieces which are then learned (the modules specialise to the subtask) and composed
> later. This seems at odds with this work’s proposed theory that how far a network deviates from the all
> gradient direction is an indicator of systematicity. ... This work, is of interest to generalisation broadly
> however, just not systematic generalisation as far as I can tell.
>
> A1. Thank you for the detailed explanation. We agree it is reasonable to require systematic generalization
> to learn internal modules or representations corresponding to the factors so that the gradient differs from
> the one computed from all data. We have the following arguments.
>
> (a) **Training with all data achieves the “effect” of systematic generalization.** The effect means
> the model has the same outputs and loss as the one with systematicity. It is a necessary condition for the
> systematic generalization. So, when a trained model has less test loss reduction, it does not achieve the
> generalization with systematicity (because the test loss is not low enough). Also, even if we only look at the
> effect, understanding the bias is still important because it would be great for standard models to achieve the
> effect.
>
> (b) **The theorem can also be applied to the cases with modules or representations.** We may
> think about a setting with the supervision of the modules or representation. We can train a model to output
> the prediction of the supervision. When all data are used in training, a model is encouraged to learn the
> correct output. So, it fits the notion of systematic generalization. It is normal training, so the theorem
> applies. The classification tasks in the experiment section correspond to such cases.
>
> Alternatively, we can consider supervision of the internal representation, though they are not usually available. It is not standard, so we do not focus on them. However, the theorem applies since the supervision
> can be a part of the loss.
>
> (c) **Why do we focus on systematic generalization?** We focus on systematic generalization because it has
> new factor combinations in test samples, so the training and the test distributions should have significant
> differences. So, it makes more sense to assume that the training and the alternative gradients do not point in
> the same direction for standard architectures. Also, systematic generalization is an important topic. Despite
> the distribution difference, systematic generalization is straightforward for humans, so it seems able to solve.
> However, it may not be true for other generalization types with such large differences.
>
> > Q3. Also the mention of ”seen test inputs” is confusing and I don’t know what this is meant to be referring
> to. But by definition test inputs are unseen.
>
> A3. It means the test input for each decomposed sub-network is seen. The whole input is unseen by
> definition. However, the input of a sub-network may be a part of the whole input, such as a convolutional
> layer or attention. It may also be a part of a hidden representation. For example, a hidden variable is a
> color, and it is invariant to other factors, even in test data.

---

> ### Author Response · Authors · 2023-11-18
> **Rebuttal 2/2**
>
> > ### Clarity
> > Q4. On the point of clarity, there are some statements in this work which do not make sense or appear
> out of context. Examples are ”Also, deep learning does not require many task-specific designs for specific
> tasks”, ”Some standard networks, such as ... work well in i.i.d. settings”, ”To keep the advantage, we discuss
> whether standard deep learning models achieve systematic generalization”.
>
> A4. These statements explain why we focus on standard models. It is because they have advantages of
> good performance and generality (apply to different tasks). So it is welcomed if they show systematic
> generalization ability.
>
> > Q5. ”The condition ∆ = 0 is rare to hold because it requires an equation to hold” and ”Both (A) and
> (B) contain equal signs, which are generally difficult to hold”. Hopefully these examples will guide a general
> clean up of the writing.
>
> A5. These cases are considered for the strictness of the theory. We will consider how to make them clearer.
>
> > Q6. A few more quicker points and concerns on clarity are the following. The last paragraph of page 2
> where the notation is introduced is also not clear and introduces more notation than necessary. Why are u
> and h defined here and why have u if x already denotes input vectors?
>
> A6. u is a vector in parameter space, so it is different from x. We study gradient, so the input is in the
> parameter space. We agree that it is more clear to define h in Definition 1, and we will make an update.
>
> > Q7. Similarly, D(f,u) is defined and includes a case for if $u = 0$ where on the top of the same page it is
> stated that $u \neq 0$.
>
> A7. Definition 1 assumes $u \neq 0$, and $u = 0$ is the extension to the definition.
>
> > Q8. ∆ is overall unhelpful as it obscures comparison with the other uses of the D(.,.) function.
>
> A8. ∆ is used to compare with zero as a threshold to discuss different cases, and it is not compared with
> other uses of D(.,.) in the main paper. We will think about how to update it.
>
> > Q9. Definitions 3 and 4 could be merged with Propositions 1, 2, and 3 since the propositions follow
> immediately from the definitions. Proposition 2 and 3 could also just be combined since the two cases are
> practically identical. DDR is also only used for two of five cases and so appears to be another function
> needlessly defined which just obscures the comparison of various cases.
>
> A9. Thank you. We will think about how to revise the paper.
>
> > Q10. Also, why not use UDDR from the beginning?
>
> A10. When the two gradients both reduce training loss, it makes sense to compare their efficiency, and we
> use DDR to compute the ratio. However, if one reduces training loss and the other does not, the ratio is
> not important because the signs are different, i.e., reducing or increasing training loss. Also, DDR has a
> straightforward interpretation, whereas UDDR may not.
>
> > Q11. The captions for Figure 3 and Table 1 should be improved and clearly state why we should care
> about these datasets, how they are used and why they relate to systematicity.
>
> A11. Thank you. We may write in the following way.
>
> Figure 3: Examples of image data. Each sample has an image and two factors: foreground and background.
> The image is the model input, and each factor is one of two classification outputs. Training data and test
> data have different factor value combinations (e.g., [cat, grass]), and each factor value is seen in training
> data. For example, the rightmost sample (d) with [bus, rock] combination is in test data, and the other
> three samples (a,b,c) are in training data. A model needs to work on unseen samples by recombining factors
> learned in training (systematic generalization), which makes learning more efficient and supports creativity.
>
> Table 1: Examples of text data. Each sample has a review (text) and two factors: category and ranking.
> Similar to the image data example, it is a systematic generalization problem, which has the review as input
> and the factors as classification outputs.

---

> > ### Comment · Reviewer_SN1k · 2023-11-22
> > **Response to Rebuttal**
> >
> > I thank the authors for their rebuttal. I will await an updated draft to assess if the clarity issues raise in Q4 to Q11 are addressed.
> >
> > Unfortunately I am unconvinced by the authors discussion in A1. The authors seem to be arguing that if a network systematically generalizes it will behave as if it has seen the full data distribution (in other words it will generalize). Ignoring more specific point on whether the network has seen all possible pieces, I agree with this. If I am correct in this interpretation then there is still a major issues. This is because the authors are then using the gradient of the full distribution to justify that the network is systematically generalizing. But this is not true. Systematic generalization (again generously ignoring details) is a sufficient condition for the network to correctly classify unseen data. Classifying unseen data is a necessary but not sufficient condition for systematic generalization. Systematic generalization is not the only way to generalize to unseen data. It is the manner in which the generalization is achieved which matters and systematic generalization is a fundamentally different manner of generalization than aligning to the potentially very entangled relationship between features and labels in the ground-truth mapping.
> >
> > Supervising modules is indeed a way to systematically generalize, but only when the modules are allocated correctly. Thus, it once again cannot be taken for granted just because a module is used [1].
> >
> > [1] Bahdanau, Dzmitry, et al. "Systematic generalization: what is required and can it be learned?." arXiv preprint arXiv:1811.12889 (2018). Finally, A3 raises other question. If I understand the implication being made here, the network sees pieces of the input during training and then must systematically generalize to the full input at test time? But this does make sense because then the notion of the network following the complete data distribution gradient would be impossible.
> >
> > Unfortunately, I will be keeping my rating the same for now.

---

> ### Author Response · Authors · 2023-11-23
>
> Thank you for your detailed comments.
>
> > A1(a). The authors seem to be arguing that if a network systematically generalizes it will behave as if
> it has seen the full data distribution (in other words it will generalize). Ignoring more specific point on
> whether the network has seen all possible pieces, I agree with this. If I am correct in this interpretation then
> there is still a major issues. This is because the authors are then using the gradient of the full distribution
> to justify that the network is systematically generalizing. But this is not true. Systematic generalization
> (again generously ignoring details) is a sufficient condition for the network to correctly classify unseen data.
> Classifying unseen data is a necessary but not sufficient condition for systematic generalization. Systematic
> generalization is not the only way to generalize to unseen data. It is the manner in which the generalization is
> achieved which matters and systematic generalization is a fundamentally different manner of generalization
> than aligning to the potentially very entangled relationship between features and labels in the ground-truth
> mapping.
>
> We would like to clarify our arguments and discuss the main concern
> in rebuttal A1(a). The point is that we argue something does not hold because its necessary condition does
> not hold.
>
> (i) We like to clarify arguments by citing a comment.
> > Classifying unseen data is a necessary but not sufficient condition for systematic generalization.
>
> The argument is that a model trained on training data does not reduce enough test loss (compared to a
> model trained on full distribution). So, it does not classify unseen data well. So, the necessary condition in
> the cited comment does not hold. So, systematic generalization (including the details) is not achieved.
>
> (ii) Discuss the main concern.
> > The authors are then using the gradient of the full distribution to justify that the network is systematically
> generalizing.
>
> In the rebuttal, we mention using the gradient of the full distribution to justify that the network (trained with the full distribution
> gradients) has the same final test loss as an ideal network that systematically generalizes (including the
> details), because they have the same test outputs. It is enough to justify the test loss because we argue
> a necessary condition does not hold (the last comment).
> *(This answer has been modified after it was initially posted.)*
>
> > A1(b). Supervising modules is indeed a way to systematically generalize, but only when the modules are
> allocated correctly. Thus, it once again cannot be taken for granted just because a module is used [1].
>
> Supervision is ground-truth, so why may the modules not be allocated correctly?

---

### Author Response · Authors · 2023-11-18
**Global response to all reviewers**

We would like to thank reviewers for their positive comments on the novelty and potential high impact of our study (SN1k,jcQf), thorough build up of clear definitions and lemmas leading to the main Theorem 1 (SN1k, ymSN), the extensive experimental results that support the main claim of the theory (SN1K, jcQf).
In the following sections, we address each reviewer's concerns separately.